# SCITAB: A Challenging Benchmark for Compositional Reasoning and Claim Verification on Scientific Tables

**Xinyuan Lu**[*1,2]   **Liangming Pan**[*3]   **Qian Liu**[4]
**Preslav Nakov**[5]   **Min-Yen Kan**[2]

[1]ISEP Program, NUS Graduate School   [2] National University of Singapore
[3]University of California, Santa Barbara   [4]Sea AI Lab   [5]MBZUAI

luxinyuan@u.nus.edu   liangmingpan@ucsb.edu   liuqian@sea.com
preslav.nakov@mbzuai.ac.ae   kanmy@comp.nus.edu.sg

## Abstract

Current scientific fact-checking benchmarks exhibit several shortcomings, such as biases arising from crowd-sourced claims and an over-reliance on text-based evidence. We present SCITAB, a challenging evaluation dataset consisting of 1.2K expert-verified scientific claims that 1) originate from authentic scientific publications and 2) require compositional reasoning for verification. The claims are paired with evidence-containing scientific tables annotated with labels. Through extensive evaluations, we demonstrate that SCITAB poses a significant challenge to state-of-the-art models, including table-based pretraining models and large language models. All models except GPT-4 achieved performance barely above random guessing. Popular prompting techniques, such as Chain-of-Thought, do not achieve much performance gains on SCITAB. Our analysis uncovers several unique challenges posed by SCITAB, including table grounding, claim ambiguity, and compositional reasoning. Our codes and data are publicly available at https://github.com/XinyuanLu00/SciTab.

## 1 Introduction

Scientific fact-checking is a crucial process that involves validating the accuracy of scientific claims by cross-referencing them with established scientific literature, research, or data (Guo et al., 2022). This process is crucial for preserving the integrity of scientific information, preventing the spread of misinformation, and fostering public trust in research findings. However, the sheer volume of scientific data and claims can be overwhelming for manual fact-checking, making automated scientific fact-checking an imperative research area of NLP.

Scientific fact-checking has advanced significantly with benchmarks including Sci-Fact (Wadden et al., 2020), Sci-Fact Open (Wadden et al., 2022), and COVID-Fact (Saakyan et al., 2021).

However, these datasets still exhibit several limitations. First, the claims are *crowd-sourced* rather than collected from real scientific papers. This leads to problems such as bias in human annotation, a lack of diversity, and shallow claims that do not reflect the complexity of scientific reasoning. For example, most claims in Sci-Fact can be validated by a single sentence in a paper's abstract, which oversimplifies the scientific discourse. Second, the claims in the existing benchmarks are solely validated against *text-based evidence*, primarily paper abstracts. However, in many scientific processes, claims are intrinsically tied to quantitative experimental data, commonly presented in tables and figures. This disparity highlights a significant gap between the existing benchmarks and real-world scientific fact-checking needs. To bridge these gaps, a dataset that 1) compiles real-world claims from scientific papers, and 2) includes original scientific data such as tables and figures, is needed.

In this paper, we propose a novel dataset SCITAB, which fulfills these stated criteria. It contains 1,225 challenging scientific claims, each demanding compositional reasoning for verification using scientific tables. Our data is derived from the Sci-Gen dataset (Moosavi et al., 2021), a resource that includes scientific tables and claims crawled from arXiv.org. We first manually filter out the check-worthy scientific claims from the raw data. Following this, we employ a strategy of human–model collaboration, as depicted in Figure 2, to generate claims that are either contradicted or unverifiable based on the table's content. Figure 1 shows a claim from SCITAB and the corresponding reasoning process to verify it. Compared with existing benchmarks, SCITAB is closer to real-world scientific fact-checking in terms of more realistic claims and table-based evidence. Through data analysis, we further show that the claims in SCITAB necessitate a more comprehensive and nuanced set of reasoning skills for verification, *e.g.,* numerical rea-

---
[*]Equal Contribution.

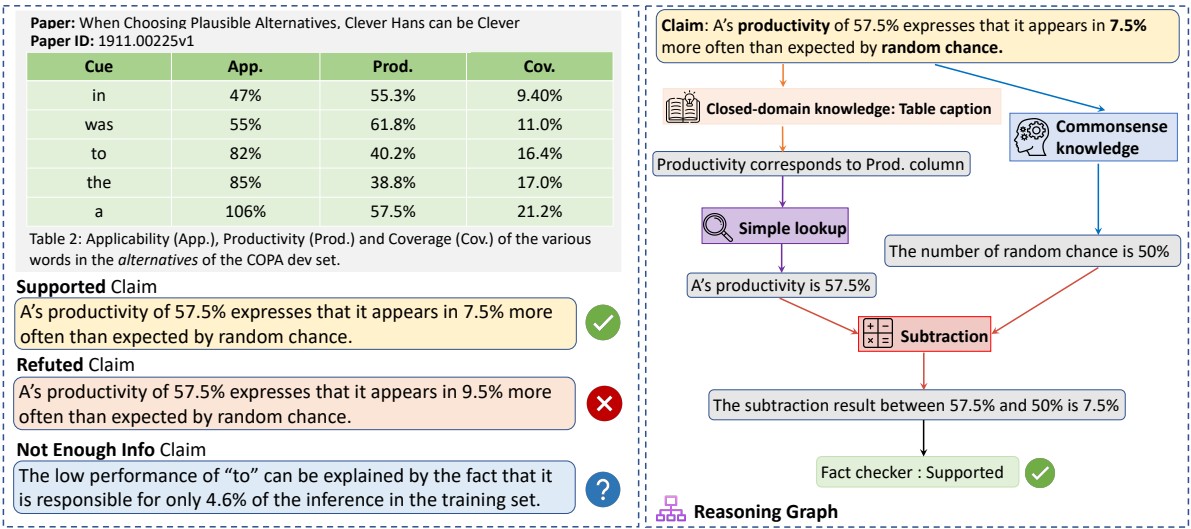

Figure 1: An example of our SCITAB dataset (left) and its corresponding reasoning graph (right). Each data entry contains *paper name*, *paper id*, *table*, one *claim*, and its corresponding *label* (Supported, Refuted, Not Enough Info).

soning and commonsense knowledge, *etc.*

We employ SCITAB as a diagnostic dataset for benchmarking the zero-shot and in-context learning performance for a wide range of state-of-the-art models, including table-based pretraining models, encoder–decoder models, open source language models, and API-based language models. We observe that all models, with the exception of GPT-4, can only achieve marginally superior $F_1$ scores than random guessing, which underscores the challenging nature of SCITAB. Additionally, established prompting methods like Chain-of-Thought (Wei et al., 2022) and Program-of-Thought (Chen et al., 2022) which typically enhance performance across most reasoning tasks, do not bring performance gain on SCITAB. Our error analysis sheds light on several unique challenges in SCITAB that may lead to this, such as table grounding, dealing with ambiguous claims, and compositional reasoning. We make our dataset fully accessible to the research community.

## 2 The SCITAB Dataset

We adopt a *human–model collaboration* strategy to construct SCITAB, as shown in Figure 2. We describe the steps involved in data preparation (Section 2.1), automatic claim generation (Section 2.2), and manual claim verification (Section 2.3).

### 2.1 Data Preparation

We use the publicly available SciGen (Moosavi et al., 2021) dataset as our primary data source.

The dataset was created by crawling computer science papers from arXiv. The tables and the texts explaining the tables are extracted from the papers to create (table, description) pairs for the task of data-to-text generation. From all the table descriptions of SciGen, we first filter the check-worthy scientific claims following the criteria established by Lee et al. (2009) for academic writing[1]. We focus on the descriptions that serve the purpose of "highlighting and commenting on key data", *i.e.*, describing research findings based on the data presented in scientific tables. Given the task's objective nature and to save the cost of human labor, we hire a graduate student majoring in computer science to manually select scientific claims based on the aforementioned criteria using the user interface in Appendix A.2. This decision was based on a pilot annotation which showed that a well-trained annotator can achieve over 95% accuracy in filtering scientific claims. To safeguard the quality, we include an option to mark the claim as "Discard-It's not a claim, or it's an incomplete, or not grammatically correct sentence." during the subsequent claim verification process. Using this approach, we filtered out 872 real-world scientific claims from 1,301 table descriptions in the SciGen dataset.

### 2.2 Automatic Claim Generation

**False Claims.** A fact-checking dataset requires both true and false claims. However, acquiring false claims that naturally occur within well-

---

[1]Detailed criteria are given in Appendix A.1

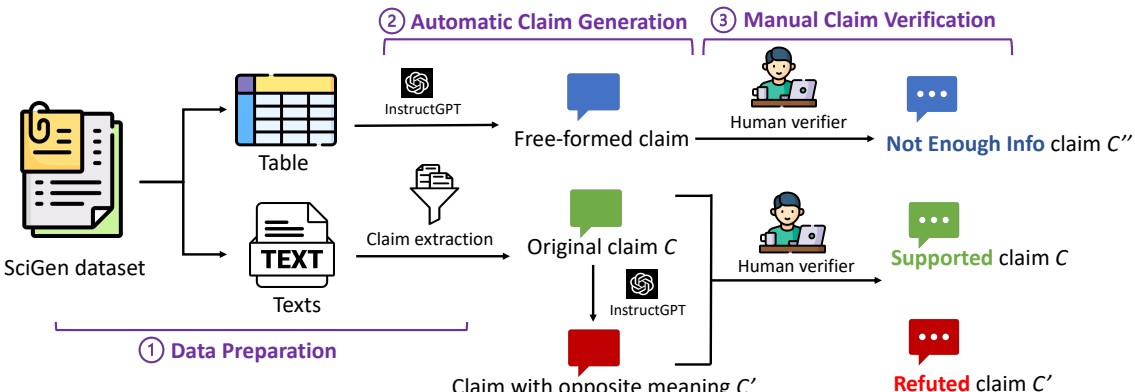

Figure 2: The *human–model collaboration* construction process of SCITAB, which contains three steps: 1) data preparation (including data preprocessing and claim extraction) 2) automatic claim generation (including refuted and Not Enough Info claim generation) and 3) manual claim verification.

verified scientific publications is a challenging task. Following SciFact (Wadden et al., 2020) and COVID-Fact (Saakyan et al., 2021), we seek to create false claims by generating counter-claims of the original true claims. Unlike previous works that purely rely on crowd-workers to compose counter-claims — a process that is costly and prone to annotation artifacts — we leverage the strong instruction-following capabilities of large language models (LLMs) to assist humans in generating candidate counter-claims. Specifically, we prompt InstructGPT (Ouyang et al., 2022) with the original claim and the instruction: *Please modify the original claims to convey the opposite meaning with minimum edits.* To foster a varied set of generated claims, we include five diverse in-context examples and employ a high decoding temperature setting of 0.7. By mandating minimal edits, we ensure that the counter-claims remain lexically close to the original claims, which is crucial in preventing fact-checking models from relying on superficial lexical patterns for verification.

**Unverifiable Claims.** To construct a more challenging dataset, we also integrate claims that are *unverifiable* with the table information (labeled as Not Enough Info, NEI). We leverage InstructGPT to generate candidate NEI claims by prompting the model with the original table and the instruction: *Please generate 5 relevant scientific claims based on the information in the table.* This process yields a diverse set of free-formed claims that enrich the diversity of SCITAB. However, as LLMs tend to generate content that might not always be grounded in the provided data, many of the generated claims turn out to be *relevant but unverifiable* with respect

to the table. We adopt manual verification (elaborated in Section 2.3) to select them as NEI claims.

## 2.3 Manual Claim Verification

We subsequently employ a human verification process for two purposes: first, to verify the quality of the 872 false claims and 900 NEI claims that were generated by InstructGPT; second, to critically review the 872 real-world scientific claims obtained in Section 2.1. This task involves selecting claims that can be verified exclusively based on the information presented in the table, without the need for additional context from the associated paper.

For each pair of the true claim $c$ and its corresponding generated counter-claim $c'$, we ask the annotator to choose one of the following three options: (A) $c$ is not exclusively supported by the table, (B) $c$ is exclusively supported by the table, but $c'$ is not refuted by the table, and (C) $c$ is not exclusively supported by the table, and $c'$ is not refuted by the table. For each candidate NEI claim, we ask the annotator to judge whether it is unverifiable with respect to the table.

**Annotator Recruitment.** Given that our data source is from computer science papers, we recruit university students majoring in computer science with basic math and programming backgrounds for annotation. We ask each annotator to fill in a questionnaire, including their age, department, maximum workload per week, etc. After that, we provide a training session to ensure they understand the task and can use the annotation interfaces (Appendix B.2 and B.3). We also give them three samples to test their understanding. We recruit twelve annotators that passed the training session.

| Statistics | | TabFact | FEVEROUS | SEM-TAB-FACTS | SCITAB |
|---|---|---|---|---|---|
| Domain | | Wiki Tables | Wiki Tables | Scientific Articles | Scientific Articles |
| Annotator | | AMT | AMT | AMT | Experts |
| Max. Reasoning Hops | | 7 | 2 | 1 | 11 |
| | Supported | 54% | 56% | 58% | 37% |
| Veracity | Refuted | 46% | 39% | 38% | 34% |
| | NEI | — | 5% | 4% | 29% |
| Total # of Claims | | 117,854 | 87,026 | 5,715 | 1,225 |
| Avg. claims per table | | 7.11 | 0.07 | 5.27 | 6.16 |

Table 1: Comparison of SCITAB to three recent table fact verification datasets: TabFact (Chen et al., 2020), FEVEROUS (Aly et al., 2021), and SEM-TAB-FACTS (Wang et al., 2021). The table presents statistics related to the domain, annotator (AMT represents Amazon Mechanical Turk), maximum reasoning hops, veracity labels percentage of each dataset, the total number of claims, and average claims per table.

In compliance with ethical guidelines, we ensure fair compensation for the annotators. Each claim annotation is reimbursed at a rate of 0.37 USD, resulting in an hourly wage of 11.2 USD[2].

**Quality Control and Annotator Agreement.** To ensure the quality of the annotation, we apply strict quality control procedures following the guidelines outlined in the Dataset Statement (Bender and Friedman, 2018). We assign two different annotators to perform a two-round annotation for each claim, while two authors review and resolve any identified errors or issues. To measure the inter-annotator agreement, we use Cohen's Kappa (Cohen, 1960). Our inter-annotator agreement is 0.630 for the false claim verification task (872 claims in total) and 0.719 for the NEI claim verification task (900 claims in total). Both values indicate substantial agreement among the annotators.

## 3 Data Analysis

Table 1 shows the statistics of our SCITAB dataset and the comparison with three existing table fact-checking datasets: TabFact (Chen et al., 2020), FEVEROUS (Aly et al., 2021), and SEM-TAB-FACTS (Wang et al., 2021). Compared with these datasets, SCITAB is 1) annotated by domain experts rather than crowd-sourced workers, 2) contains more challenging claims that require up to 11 reasoning steps for verification, and 3) has a more balanced distribution of veracity labels and a higher percentage of NEI claims. We conduct a more in-depth analysis of SCITAB as follows.

### 3.1 Reasoning Analysis

**Reasoning Types.** To study the nature of reasoning involved in fact-checking claims in SCITAB,

we adapt the set of table-based reasoning categories from INFOTABS (Gupta et al., 2020) to define 14 atomic reasoning types, as shown in Table 2. Among them, "closed-domain knowledge" and "open-domain knowledge" are specially designed for SCITAB. Closed-domain knowledge refers to obtaining background information from the table caption or title, *e.g.*, knowing that "Prod." refers to "Productivity" from the table caption in Figure 1. Open-domain knowledge refers to commonsense knowledge not presented in the table, *e.g.*, the relationship between precision and recall. Given the designed reasoning types, we manually analyze 100 samples in SCITAB, by annotating the graph of reasoning steps for verifying each claim. We identify 476 atomic reasoning steps from the 100 analyzed samples and show the proportion for each reasoning type in Table 2. We observe that SCITAB has a multifaceted complex range of reasoning types and a high proportion of claims requiring different types of domain knowledge.

**Reasoning Depth.** We further measure the *reasoning depth* (the number of required reasoning steps) for each claim and show the reasoning depth distribution in Figure 3. We find that the analyzed claims have an average depth of 4.76 and a maximum depth of 11. Moreover, 86% of the claims requiring 3 or more reasoning steps, which demonstrates the complexity of reasoning in SCITAB.

**Reasoning Graph.** We showcase the reasoning graph for the example in Figure 1 on the right side of the figure. Verifying this claim requires various types of reasoning including: 1) *background knowledge from the table caption*: "productivity" corresponds to the "Prod." column in the table; 2) *commonsense knowledge*: "random chance" means 50% accuracy; 3) *simple lookup*: "A's productivity" refers to the cell located at the last row and

---

[2] The payment is fair and aligned with the guideline for dataset creation (Bender and Friedman, 2018).

| Function Names | Descriptions | Prop. (%) |
|---|---|---|
| Simple lookup | Retrieve the value for a specific cell. | 20.6 |
| Comparison | Compare two numbers. | 19.5 |
| Closed-domain knowledge | Extract information from context sentences in the table caption or article. | 12.1 |
| Open-domain knowledge | Extract additional information required by domain experts. | 5.3 |
| Commonsense knowledge | Extract commonsense knowledge necessary for claim verification. | 5.3 |
| Subtract | Perform subtraction of two numbers. | 5.3 |
| Divide | Perform division of two numbers. | 5.3 |
| Rank | Determine the rank of a set of numbers. | 5.3 |
| Different / Same | Determine if two numbers are different or the same. | 5.3 |
| Add | Calculate the sum of two numbers. | 4.0 |
| Max / Min | Retrieve the maximum or minimum number from a set of numbers. | 3.1 |
| Col / Rowname | Retrieve the column or row name from the table. | 3.1 |
| Trend same/different | Determine the trend for two columns or rows, whether they are the same or different. | 2.9 |
| Set check | Verify if a value belongs to a set of numbers. | 2.9 |

Table 2: The function names, descriptions, and their proportions in our SCITAB dataset.

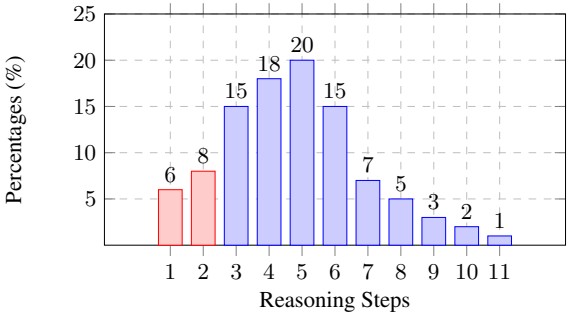

Figure 3: The distribution histogram of reasoning steps in our SCITAB dataset. The x-axis is the reasoning steps in each claim, and the y-axis is the frequency for each reasoning step. The shallow claims (with 1–2 reasoning steps) are highlighted in red, while the deep claims (with 3+ reasoning steps) are highlighted in blue.

the "Prod." column; and 4) *numerical reasoning*: the difference between 57.5% and 50% is 7.5%. This case study provides further insights into the complexity and variety of reasoning involved in SCITAB, revealing the difficulty of the dataset.

### 3.2 Refuted and NEI Claims Analysis

One potential risk of model-generated claims is that they may lack diversity and exhibit the same pattern. For example, in the Sci-Fact (Wadden et al., 2020) dataset where the refuted claims are generated by flapping the meaning of the original true claims, we found that out of 100 randomly sampled refuted claims, 85 simply negated the original claim by adding negation words such as "not" (more details in Appendix C). To evaluate the diversity of claims for our SCITAB dataset, we randomly select 60 refuted claims and then manually annotate their reasons for refutation. Results are shown in Table 3 (top half). We find that SCITAB exhibits

a greater diversity in refuted claims compared to Sci-Fact. Besides common error types such as "incorrect calculation results" (41.7%), there are also unique types of errors that are more reflective of the complexities in real-world scientific claims. For example, 33.33% of the refuted claims contain "incorrect approximation words", and 10.0% are cases where "the claim is partially right", consistent with the fact that ambiguity and half-truths are common phenomena in scientific discourse. Additional examples of refuted claims are in Appendix E.

The NEI claims (bottom half; Table 3) also exhibit diverse reasoning patterns. The two most common features for unverifiable claims are insufficient evidence in the table and the lack of background knowledge. The lack of closed-domain knowledge is another reason for NEI, where additional information in the paper is necessary to verify the claim. Other reasons include the use of vague pronouns (*e.g.*, "it", "this") brings ambiguity to the claim. These distinct refuted and NEI reasoning types highlight the unique features of SCITAB, making it a more comprehensive and realistic representation of the challenges faced in real-world scientific fact-checking.

## 4 Experiment

We formally define the task of scientific table-based fact-checking as follows. A scientific table $\mathcal{T}$ consists of a table caption $P$ and the table content ($\{T_{i,j} | i \leq R_T, j \leq C_T\}$ with $R_T$ rows and $C_T$ columns, where $T_{i,j}$ is the content in the $(i,j)$th cell. Given a claim $C$ describing a fact to be verified against the table $\mathcal{T}$, a *table fact-checking* model $\mathcal{F}$ predicts a label $\mathcal{Y}$ to verify whether $\mathcal{C}$ is supported, refuted, or can not be verified by the

| Refuted Reasons | Prop. (%) |
|---|---|
| The calculation result is wrong. | 41.7 |
| The approximation word is wrong. | 33.3 |
| The claim is partially right. | 10.0 |
| The values in the claim do not match. | 8.3 |
| The operation type is wrong. | 6.7 |
| **NEI Reasons** | **Prop. (%)** |
| The claim does not have enough matching evidence. | 33.3 |
| The claim lacks open-domain knowledge. | 25.0 |
| The claim lacks closed-domain knowledge. | 15.0 |
| The claim refers to another table. | 11.7 |
| The claim contains vague pronouns. | 8.3 |
| The claim omits specific information. | 6.7 |

Table 3: Refuted and NEI reasons and their estimated proportions (Prop.) in SCITAB.

information in $\mathcal{T}$.

Considering the real-world situation that large-scale training data is either not available or expensive to collect, we focus on the *zero-shot/in-context* evaluation where the model can only access zero/few in-domain data from SCITAB. To this end, we randomly hold out 5 tables with 25 claims as model-accessible data and use the rest of the data as the unseen test set. This also prevents the model from learning spurious features that lead to over-estimated performance (Schuster et al., 2019).

## 4.1 Models

We conduct a comprehensive evaluation of SCITAB for various models, including table-based pretraining models, encoder–decoder models, open source LLMs, and closed source LLMs. We also study the human performance to analyze the upper bounds on SCITAB.

**Table-based LLMs.** These are pre-trained transformer models fine-tuned on tabular data. We choose three different models: 1) TAPAS (Herzig et al., 2020), a BERT-based model fine-tuned on millions of tables from English Wikipedia and corresponding texts, 2) TAPEX (Liu et al., 2022b), a model that fine-tunes BART (Lewis et al., 2020) on a large-scale synthetic dataset generated by synthesizing executable SQL queries and their execution outputs, and 3) TAPEX-Zero (Liu et al., 2023b), an enlarged version of TAPEX. For TAPAS and TAPEX, we use their fine-tuned version on TabFact (Chen et al., 2020) for table fact-checking.

**Encoder–Decoder LLMs.** We also use encoder–decoder models where both the input and output are sequences of tokens. To adapt the model to take the table as input, we flatten the table as a sequence following Chen et al. (2020). The in-

put is then formulated as $[\tilde{T}; P; C; Q]$, where $\tilde{T}$ is the linearized table, and $Q$ is a question template "Based on the information in the table, is the above claim true? A) True B) False C) Unknown?". We choose FLAN-T5 (Chung et al., 2022), an improved T5 model (Raffel et al., 2020) pre-trained on more than 1.8K tasks with instruction tuning, which has achieved strong zero-shot/in-context performance on other fact-checking benchmarks.

**Open Source LLMs.** We also evaluate the performance of state-of-the-art open source LLMs, including 1) LLaMA (Touvron et al., 2023), the first open-source model by Meta AI; 2) Alpaca (Taori et al., 2023), an instruction-following language model fine-tuned on LLaMA; and 3) Vicuna (Chiang et al., 2023), the arguably best-performed open-source LLMs that claimed to achieve 90% quality compared to OpenAI ChatGPT. We use the same input format as in the encoder-decoder model.

**Closed Source LLMs.** These are closed-source LLMs that require API calls for inference, including InstructGPT (text-davinci-003) (Ouyang et al., 2022) and GPT-4 (OpenAI, 2023). We evaluate the setting that directly predicts the label and the Chain-of-Thought (CoT) (Wei et al., 2022) setting, which generates explanations before predicting the final label. We also include the Program-of-Thoughts (PoT) (Chen et al., 2022) model that has shown strong ability in solving complex numerical reasoning tasks. It first parses the reasoning steps as Python programs and then executes them on a Python interpreter to derive accurate answers. Since most claims in SCITAB also require numerical reasoning, we want to test whether program-guided reasoning can be extended to table-based fact-checking.

**Human Performance.** To examine how humans perform on our SCITAB dataset, we hired an annotator from our candidate annotators pool, following the same training procedure as other annotators. In the case of 2-class classification, we randomly selected 40 samples: 20 each for supported and refuted claims. For 3-class classification, we randomly selected 60 random samples, ensuring an even distribution of 20 samples across the three label categories (supported, refuted, and not enough information). The annotator took approximately 1.5 hours for the 2-class fact-checking task and 2 hours for the 3-class setting. We report the Macro-F1 scores at the bottom of Table 4.

| | Models | # of Para. | Zero-shot | | In-Context | |
|---|---|---|---|---|---|---|
| | | | 2-class | 3-class | 2-class | 3-class |
| I. Table-based LLMs | TAPAS-large (Tabfact) (Herzig et al., 2020) | 340M | 50.30 | — | — | — |
| | TAPEX-large (Tabfact) (Liu et al., 2022b) | 400M | 56.06 | — | — | — |
| | TAPEX-Zero-large (Liu et al., 2023b) | 780M | 48.28 | 29.72 | 42.44 | 23.47 |
| | TAPEX-Zero-XL (Liu et al., 2023b) | 3B | 49.77 | 34.30 | 42.12 | 25.62 |
| II. Encoder–Decoder LLMs | Flan-T5-base (Chung et al., 2022) | 250M | 47.38 | 26.56 | 44.82 | 24.09 |
| | Flan-T5-large (Chung et al., 2022) | 780M | 51.58 | 32.55 | 49.62 | 27.30 |
| | FLan-T5-XL (Chung et al., 2022) | 3B | 52.41 | **38.05** | 48.05 | 29.21 |
| | Flan-T5-XXL (Chung et al., 2022) | 11B | 59.60 | 34.91 | **60.48** | 34.04 |
| III. Open source LLMs | Alpaca-7B (Taori et al., 2023) | 7B | 37.22 | 27.59 | 40.46 | 28.95 |
| | Vicuna-7B (Chiang et al., 2023) | 7B | **63.62** | 32.47 | 50.35 | 34.26 |
| | Vicuna-13B (Chiang et al., 2023) | 13B | 41.82 | 29.63 | 55.11 | **35.16** |
| | LLaMA-7B (Touvron et al., 2023) | 7B | 49.05 | 32.26 | 45.24 | 27.17 |
| | LLaMA-13B (Touvron et al., 2023) | 13B | 53.97 | 37.18 | 44.39 | 32.66 |
| IV. Close source LLMs | InstructGPT (Ouyang et al., 2022) | 175B | 68.44 | 41.41 | 68.10 | 41.58 |
| | InstructGPT+CoT (Ouyang et al., 2022) | 175B | — | — | 68.46 | 42.60 |
| | PoT (Chen et al., 2022) | 175B | — | — | 63.79 | — |
| | GPT-4 (OpenAI, 2023) | — | 78.22 | 64.80 | 77.98 | 63.21 |
| | GPT-4+CoT (OpenAI, 2023) | — | — | — | 76.85 | 62.77 |
| | Human | — | — | — | 92.40 | 84.73 |

Table 4: Macro-$F_1$ of baselines on SCITAB for different settings. The *# of Para.* indicates the number of parameters in the models. The TAPAS and TAPEX models are fine-tuned on the TabFact dataset, while others perform zero-shot learning. The bold text indicates the best performance among I to III, while the underlined text indicates the overall best performance among all the models.

## 4.2 Main Results

We evaluate all models under both *zero-shot* and *in-context* settings. In the zero-shot setting, the model does not have access to any in-domain data. In the in-context setting, we provide three hold-out examples as demonstrations. We report two sets of results: the *2-class* case, where examples labeled as NEI are excluded (since some models cannot process NEI claims), and the *3-class* case including all three labels. The results are shown in Table 4. We have five major observations.

1. In general, all open source LLMs, including encoder–decoder models and decoder-only models, do not achieve very promising results on SCITAB and they still have a large gap from human performance. The best result is 63.62 for the 2-class setting (Vicuna-7B and 38.05 for the 3-class setting (FLAN-T5-XL). Both results are only moderately better (+13.62 and +4.72) than random guessing. In contrast, a well-trained human annotator can achieve 92.46 and 84.73 F1 scores in the 2-class and 3-class settings, respectively. This reveals the challenging nature of SCITAB and its potential to be the future benchmark for scientific fact-checking.

2. Counter-intuitively, table-based LLMs do not outperform models pre-trained on pure texts, for example, FLAN-T5. This discrepancy may be attributed to the dissimilarity between the distribution of tables in scientific literature and publicly available table corpus. For example, scientific tables commonly include both row and column headers, whereas most tables in Wikipedia lack row headers. Meanwhile, the claims in our dataset are usually much longer than those in previous works, raising challenges to table-based LLMs.

3. The results in the 3-class setting are notably poorer than those in the 2-class setting. This discrepancy reveals the challenges that most models face when confronted with the NEI class. One plausible explanation could be the inherent difficulty in distinguishing between 'refuted' and 'NEI' claims — a task that even trained human annotators struggle with, as noted by Jiang et al. (2020). Our forthcoming error analysis will further demonstrate that the inclusion of the NEI class tends to diminish the models' confidence, causing a shift in their predictions from 'supported/refuted' to 'NEI'.

4. Interestingly, the provision of in-context examples does not result in improved performance for

the majority of models. This observation is somewhat expected for open source LLMs as they have not been reported to possess in-context learning capabilities. Nonetheless, it is surprising to find that even with chain-of-thought prompting, in-context demonstrations do not yield positive effects for InstructGPT and GPT-4. Our error analysis on the PoT offers some insight into this phenomenon and will be discussed in the next section.

5. Closed source LLMs perform better than open source LLMs, with GPT-4 achieving 78.22 macro-$F_1$ for the 2-class setting and 64.80 for the 3-class setting. This aligns with the assertion that GPT-4 has a strong ability to perform complex reasoning (OpenAI, 2023) and we show that this ability can generalize to tabular data as well. However, the black-box nature of OpenAI models restricts our further analysis of its behavior.

### 4.3 Error Analysis

**InstructGPT and GPT-4.** We show the confusion matrices for InstructGPT and GPT-4 under the zero-shot 3-class setting in Figure 4. We find that both models have difficulty in accurately predicting the NEI class. InstructGPT displays a pattern of "less confident", frequently classifying supported and refuted claims as 'NEI'. In contrast, GPT-4 exhibits overconfidence, incorrectly categorizing NEI claims as either supported or refuted. This corroborates our earlier observation that distinguishing whether a claim is *verifiable* is one of the key challenges for SCITAB.

Further, we also examine individual error instances, with typical examples provided in Figures 11 and 12 of Appendix F. The majority of 'supported' claims that were incorrectly classified as 'refuted' (Case 6) involve numerical reasoning or comparison. Conversely, when 'refuted' claims are inaccurately predicted as 'supported' (Case 3), we find that LLMs often overlook claims containing negation, indicating a lack of deep comprehension. For cases where 'supported' or 'refuted' claims are erroneously predicted as 'NEI' (Cases 1 and 2), such claims typically demand extensive reasoning and a deep understanding of the research findings. Interestingly, when faced with these complex cases, the model tends to default to the safer choice of 'uncertain' (NEI).

**PoT.** Unexpectedly, incorporating a Python interpreter does not confer any advantage on our dataset (as shown in Table 4), despite its positive

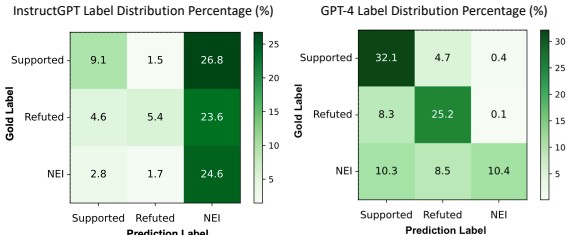

Figure 4: Confusion matrix for InstructGPT (left) and GPT-4 (right) in the zero-shot 3-class classification task.

| Error Type | Estimated Proportion (%) |
|---|---|
| I. Grounding errors | 50 |
| II. Ambiguity errors | 22 |
| III. Calculation errors | 20 |
| IV. Program errors | 8 |

Table 5: The error types and their estimated proportions for incorrectly-predicted samples in PoT.

impacts on other numerical reasoning tasks. In order to understand this, we randomly selected 50 claims wherein the PoT incorrectly predicted the final veracity labels and evaluated the quality of the generated Python programs. We divide the errors into four categories, as assessed by human annotators: (*i*) *Grounding errors*, where the program incorrectly associates data with the respective cells in the table; (*ii*) *Ambiguity errors*, where the claim contains ambiguous expressions that the program fails to represent; (*iii*) *Calculation errors*, where incorrect floating point arithmetic calculation in Python lead to inaccurate results and (*iv*) *Program errors*, which encompass mistakes such as incorrect or missing arguments/variables, and erroneous operations. We present the error analysis in Table 5, and examples of program errors can be found in Figure 13 and Figure 14 in Appendix G. Compared to other datasets, categories (*i*) and (*ii*) present unique challenges in our dataset. Category (*i*) underlines the difficulty in accurately referencing the specific cells to which a claim refers. Category (*ii*), on the other hand, emphasizes the difficulties posed by the ambiguous nature of scientific claims, such as "A is significantly better than B", to program-based methods. This connection further emphasizes the contribution of our work in addressing the mismatches between reasoning types and the occurrence of grounding errors.

## 5 Related Work

**Scientific Fact-Checking Datasets.** Existing datasets for scientific fact-checking are summa-

rized in a recent survey from Vladika and Matthes (2023). These datasets differ in: 1) *domain*: biology (Wadden et al., 2020; Akhtar et al., 2022), COVID-19 (Saakyan et al., 2021; Sarrouti et al., 2021; Mohr et al., 2022; Wang et al., 2023), and climate (Diggelmann et al., 2020), 2) *claim creation*: crowd-sourced claims *v.s.* natural claims, and 3) *evidence source*: Wikipedia articles (Diggelmann et al., 2020) or research papers (Wadden et al., 2020, 2022; Sarrouti et al., 2021). However, most of these datasets rely on text evidence to verify claims. SEM-TAB-FACTS (Wang et al., 2021) is the only existing dataset based on scientific tables, but it is limited to simple, crowd-sourced claims. To bridge this gap, we construct SCITAB which contains complex claims from authentic scientific papers with table-based evidence.

**Table-based Reasoning.** Table-based reasoning requires reasoning over both free-form natural language queries and (semi-)structured tables. Early works either rely on executable languages (*e.g.*, SQL and SPARQL) to access the tabular data (Yin et al., 2016; Yu et al., 2018) or employ graph neural networks to capture logical structure in statements, *e.g.*, LogicFactChecker (Zhong et al., 2020) and ProgVGAT (Yang et al., 2020). However, these approaches often struggle with generalization, as they are tightly bound to specific table formats and language patterns. To address this, we have seen a shift toward table pre-training, with the advent of Table-BERT (Chen et al., 2020), TAPAS (Herzig et al., 2020), SaMoE (Zhou et al., 2022), PASTA (Gu et al., 2022), and DATER (Ye et al., 2023). These methods encode sentence-table pairs using language models and transform table-based reasoning into question-answering or natural language inference. In our work, we focus on evaluating pre-training-based methods on SCITAB because they not only demonstrate superior performance but also offer the benefits of few-shot learning.

## 6 Conclusion and Future Work

We present SCITAB, a novel dataset for scientific fact-checking that addresses the limitations of existing benchmarks. By incorporating real-world scientific claims and their corresponding evidence in the form of tables, SCITAB offers a more comprehensive and fine-grained representation of scientific reasoning. The challenging nature of SCITAB is evident from the performance of the state-of-the-art, highlighting the need for further

research. For example, we believe that addressing the challenges posed by ambiguous claims represents a crucial direction for research in scientific fact-checking (Glockner et al., 2023; Liu et al., 2023a). One potential approach is to enhance the disambiguation of ambiguous claims by leveraging contextual information or external knowledge sources. Additionally, studying the compositionality in table-based reasoning is an interesting direction. Consider the work of Self-Ask (Press et al., 2022), which proposed the "compositionality gap" metric to measure the capability of LLMs in compositional reasoning. Such evaluations can be enriched by annotating SCITAB with ground-truth reasoning depths and structured reasoning graphs. Beyond this, another direction worth exploring is equipping the LLMs with external tools to further improve the model. For example, the use of GPT-4 plugins, Program-guided Fact-Checking (Pan et al., 2023) or adopting approaches from other tool-augmented LLMs like Toolformer (Schick et al., 2023) and Chameleon (Lu et al., 2023).

## Ethics Statement

We have received approval from the Institutional Review Board (IRB)[3] for our data collection. The IRB reviewed our experimental design and research procedures to ensure that they do not pose more than minimal risks to research participants. We take steps to protect research participants' privacy and the confidentiality of their data. The review process took two months to complete.

## Limitations

Firstly, the method and dataset are primarily designed for languages with limited morphology, such as English. Secondly, our SCITAB dataset is specifically focused on fact-checking scientific claims based on tables, which represents only one aspect of scientific fact-checking. Further research can explore the integration of other forms of evidence, including textual evidence and figure evidence, to enhance the fact-checking process. Thirdly, our SCITAB dataset is primarily focused on numerical reasoning types, as it is derived from the SciGen dataset, which also emphasizes numerical reasoning. It would be beneficial for future studies to incorporate a wider range of reasoning types to provide a more comprehensive fact-checking

---

[3]https://www.nus.edu.sg/research/irb. The NUS-IRB Reference Code is NUS-IRB-2022-599

framework. Lastly, it would be valuable to explore additional annotation types, such as reasoning graphs, to further enrich the depth of analysis and capture more intricate relationships within the claims and evidence.

## Acknowledgements

This research is supported by the Ministry of Education, Singapore, under its MOE AcRF Tier 3 Grant (MOE-MOET32022-0001). The computational work for this article was partially performed on resources of the National Supercomputing Centre, Singapore (`https://www.nscc.sg`).

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

## A  Claim Extraction Procedure

### A.1  Claim Definition

In academic writing (Lee et al., 2009), the accompanying text for data, presented as tables and figures), typically includes three fundamental elements as outlined below. These elements encompass the definition of claims, which involve highlighting key data (KD) and commenting on key data (COM) that emphasizes and comments on the key data.

**Location of results (LOC).**  Statements that locate where the figure/table is found, *e.g.,* Figure 7 displays the mean percentile scores.

**Highlighting of key data (KD).**  Statements that highlight the important data, *e.g.,*(1) Highest or lowest values (2) Overall trend or pattern in the data (3) Points that do not seem to fit the pattern or trend, etc. (4) Results which provide answers to your research questions

**Commenting on key data (COM).**  Statements that interpret the data. There are three types of comments: (1) Generalization (deductions and implications drawn from the results), *e.g.,* "This indicates that ..." (2) Comparison of results with those from prior studies, *e.g.,* "Different from ..." (3) Explanation or speculation (possible reasons or cause-effect relationships for the results), *e.g.,* "The possible reason is that ..."

### A.2  Claim Extraction Interface

Figure 5 shows the user interface for the claim extraction task.

## B  Manual Claim Verification Procedure

### B.1  Annotator Training Process

Our annotator selection and training process is systematic and thorough to ensure the highest quality annotations. We initiate the process by advertising on our university's platform. Interested candidates are then required to complete a registration form. From these responses, the authors identify suitable annotators based on set criteria. Once shortlisted, the potential annotators are invited for a training session, which can be conducted either in-person or via Zoom, lasting approximately one hour. This session is divided into three parts. Firstly, the authors provide a comprehensive overview of the task definition, ensuring clarity on what is expected. Similar to WANLI (Liu et al., 2022a), during our

training sessions[4], commonsense interpretations and a minimum amount of logical inference are acceptable. Next, a demonstration is given on how to navigate and utilize the annotation interface effectively. Following this, a series of trial tests are released to the annotators. This is to verify their understanding and capability in the task. Last, we specify the deadline for completing annotations, outline how we check the quality of their work, brief them on a post-annotation survey, and explain the reimbursement procedure. A Q&A session is also incorporated to address any uncertainties or concerns. After receiving their reimbursement, the annotators signed an agreement sheet to ensure its receipt.

### B.2  NEI Claim Verification Interface

Figure 6 shows the user interface for the NEI claim verification task.

### B.3  Refuted Claim Verification Interface

Figure 7 shows the user interface for the refuted claim verification task.

### B.4  Annotation Post-Survey

Figure 8 shows the examples of post-annotation survey questions and the answers of annotators.

## C  Analysis of Refuted Reasons in the Sci-Fact dataset

Table 6 provides an analysis of the reasons for refuted claims in the Sci-Fact dataset, along with their estimated proportions. A random sample of 100 refuted claims was selected, and the results indicate that 85% of claims were simply negated using terms like "not" or paraphrased based on the evidence sentences. Additionally, 6% of the refuted claims were attributed to incorrect calculation results, while 6% were identified as having wrong commonsense knowledge. A smaller proportion of refuted claims (3%) were found to have incorrect open-domain knowledge.

## D  Discussions on Human-Machine Collaboration

Our final data creation pipeline undergoes repetitive testing and revision until it reaches its current

---

[4]All the related materials including the advertisement, a sample of the registration form and the agreement sheet are available at `https://github.com/XinyuanLu00/SciTab`.

| Refuted Reasons | Prop. (%) |
|---|---|
| Negation (+not) and paraphrasing. | 85 |
| The calculation result is wrong. | 6 |
| The commonsense knowledge is wrong. | 6 |
| The open-domain knowledge is wrong. | 3 |

Table 6: The refuted reasons and their estimated proportions (Prop.) in the Sci-Fact dataset.

form. In our pilot annotation, we found that manual verification played the most essential role in the validation of claims marked as "Not Enough Information(NEI)". Initially, we planned to rely solely on LLMs for generating NEI claims. Our criteria for the NEI claim is that "the claim should be fluent, logical, and relevant to the table. However, the claim cannot be verified as true or false solely based on the information in the table." However, after a careful examination of the LLM output, we found that LLM tends to generate claims that are either not logical or irrelevant to the table content. Therefore, human efforts are required to further select NEI claims that meet our criteria. Out of 900 initial NEI claims generated by LLMs, manual verification narrowed them down to only 355 claims, taking up 40% of the original count. While it may not have served as crucial a role as filtering NEI claims, human verification also safeguarded the data quality in other annotation processes. For example, among the "supported" claims originally appearing in the scientific paper, human validation still identified 10 cases that were actually not supported (*e.g.*, wrong number matching.)

## E   Case Study for Refuted Claims

Figure 9 and Figure 10 show five examples of refuted cases. Below, we provide explanations for each of these error cases.

**Case *A* The calculation result is wrong.**   It produces incorrect calculation results. The accurate result should be `27.9-21.7 = 6.2`.

**Case *B* The approximation word is wrong.**   It generates incorrect approximation words, as 19.4 is not significantly lower compared to 23.3.

**Case *C* The claim is partially right.**   The claim is generally correct, with the exception of the `BShift` column which does not fulfill the claim.

**Case *D* The values in the claim do not match.** The value in the claim does not align with the cor-

responding value in the table. The correct value should be `27.9`.

**Case *E* The operation type is wrong.**   It applies the incorrect operation type. For instance, in the case of `GCN+RC+LA (9)`, it is not accurate to claim that it is better than `DCGCN1` because `22.9 > 22.0` and `53.0 > 52.6`.

## F   Error Cases for InstructGPT

Figure 11 and Figure 12 show six error examples of InstructGPT in the zero-shot setting when applied to our SCITAB dataset.

**Error Type 1: Supported predicted as NEI.** This error type indicates a discrepancy between the gold label, which is Supported, and the predicted label, which is NEI.

**Error Type 2: Refuted predicted as NEI.**   This error type indicates a discrepancy between the gold label, which is Refuted, and the predicted label, which is NEI.

**Error Type 3: Refuted predicted as Supported.** This error type indicates a discrepancy between the gold label, which is Refuted, and the predicted label, which is Supported.

**Error Type 4: NEI predicted as Supported.** This error type indicates a discrepancy between the gold label, which is NEI, and the predicted label, which is Supported.

**Error Type 5: NEI predicted as Refuted.**   This error type indicates a discrepancy between the gold label, which is NEI, and the predicted label, which is Refuted.

**Error Type 6: Supported predicted as Refuted.** This error type indicates a discrepancy between the gold label, which is Supported, and the predicted label, which is Refuted.

## G   Error Cases for Program-of-Thoughts

Figure 13 and Figure 14 show five error examples of Program-of-Thoughts when applied to our SCITAB dataset. Below, we provide explanations for each of the error cases.

**Error Case 1.**   It exhibits incorrect entity linking (*Grounding error*) and incorrect operation (*Program error*). The codes "`winograd_baseline = 73.06`" and "`winocoref_baseline = 88.48`" should be "`IlliCons_winograd = 53.26`" and

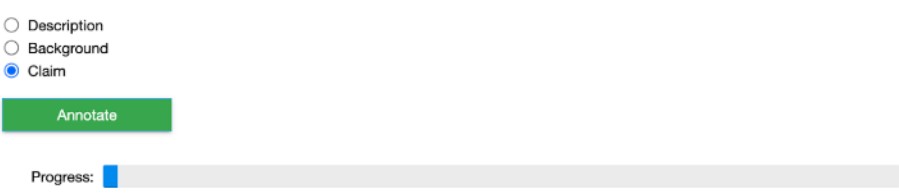

**ID:** 2-2

**Paper ID:** 1805.11461v1

**Paper Name:** Syntactic Dependency Representations in Neural Relation Classification

| Representation | Hyper parameters Filter size | Hyper parameters Num. Feature maps | Hyper parameters Activation func. | Hyper parameters L2 Reg. | Hyper parameters Learning rate | Hyper parameters Dropout Prob. | F1.(avg. in 5-fold) with default values | F1.(avg. in 5-fold) with optimal values |
|---|---|---|---|---|---|---|---|---|
| CoNLL08 | 4-5 | 1000 | Softplus | 1.15e+01 | 1.13e-03 | 1 | 73.34 | 74.49 |
| SB | 4-5 | 806 | Sigmoid | 8.13e-02 | 1.79e-03 | 0.87 | 72.83 | 75.05 |
| UD v1.3 | 5 | 716 | Softplus | 1.66e+00 | 9.63E-04 | 1 | 68.93 | 69.57 |

**Caption:** Table 2: Hyper parameter optimization results for each model with different representation. The max pooling strategy consistently performs better in all model variations.

**Sentence to annotate:**

The results furthermore show that the sdps based on the Stanford Basic (SB) representation provide the best performance, followed by the CoNLL08 representation.

○ Description
○ Background
● Claim

[ Annotate ]

Progress: ▮

Figure 5: The user interface for the claim extraction task.

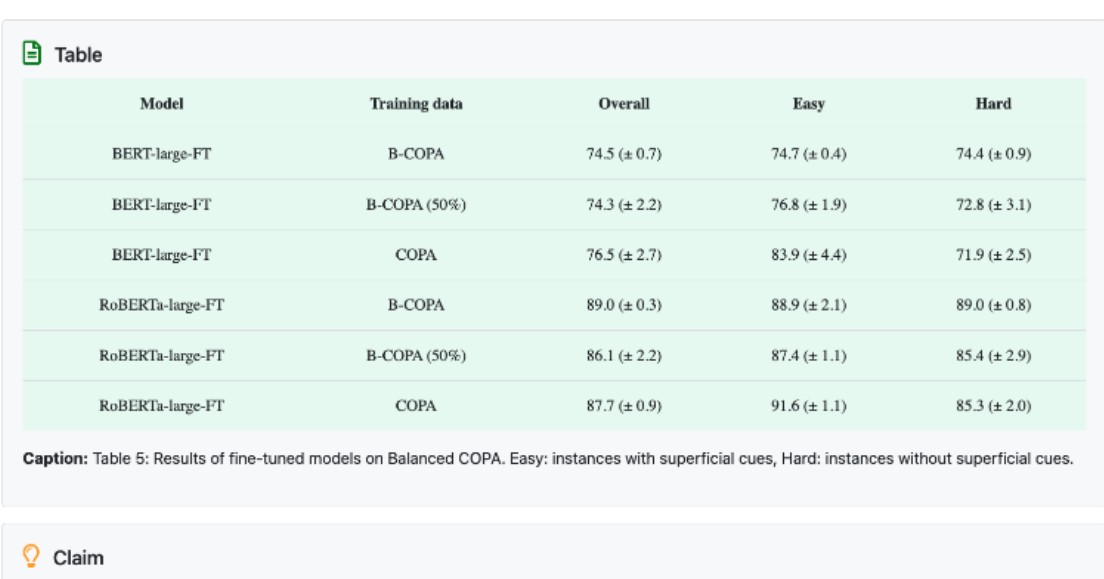

**ID:** 0-0-0

**Paper ID:** 1911.00225v1

**Paper:** When Choosing Plausible Alternatives, Clever Hans can be Clever

🗎 Table

| Model | Training data | Overall | Easy | Hard |
|---|---|---|---|---|
| BERT-large-FT | B-COPA | 74.5 (± 0.7) | 74.7 (± 0.4) | 74.4 (± 0.9) |
| BERT-large-FT | B-COPA (50%) | 74.3 (± 2.2) | 76.8 (± 1.9) | 72.8 (± 3.1) |
| BERT-large-FT | COPA | 76.5 (± 2.7) | 83.9 (± 4.4) | 71.9 (± 2.5) |
| RoBERTa-large-FT | B-COPA | 89.0 (± 0.3) | 88.9 (± 2.1) | 89.0 (± 0.8) |
| RoBERTa-large-FT | B-COPA (50%) | 86.1 (± 2.2) | 87.4 (± 1.1) | 85.4 (± 2.9) |
| RoBERTa-large-FT | COPA | 87.7 (± 0.9) | 91.6 (± 1.1) | 85.3 (± 2.0) |

**Caption:** Table 5: Results of fine-tuned models on Balanced COPA. Easy: instances with superficial cues, Hard: instances without superficial cues.

💡 Claim

For both BERT and RoBERTa, more data leads to more performance improvements, although the growth rate on RoBERTa is slightly faster.

Figure 6: The user interface for the NEI claim verification task.

"IlliCons_winocoref = 74.32" respectively. Additionally, the ">" operation should be changed to ">=".

**Error Case 2.** It exhibits incomplete entity linking (*Grounding error*). The program should also parse other baseline results, such as 'SFEGAN_WER = 14.9".

**Error Case 3.** It fails to generate a correct program (*Program error*). The variables and logical functions in the programs are incorrect. For in-

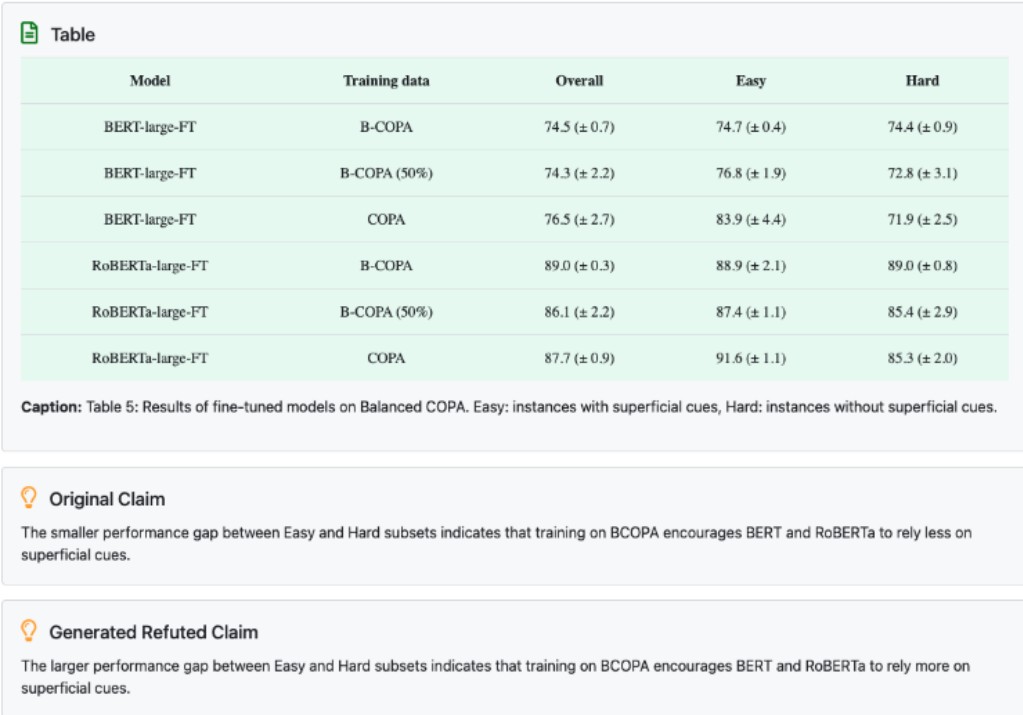

ID: train-0-1

Paper ID: 1911.00225v1

Paper: When Choosing Plausible Alternatives, Clever Hans can be Clever

**Table**

| Model | Training data | Overall | Easy | Hard |
|---|---|---|---|---|
| BERT-large-FT | B-COPA | 74.5 (± 0.7) | 74.7 (± 0.4) | 74.4 (± 0.9) |
| BERT-large-FT | B-COPA (50%) | 74.3 (± 2.2) | 76.8 (± 1.9) | 72.8 (± 3.1) |
| BERT-large-FT | COPA | 76.5 (± 2.7) | 83.9 (± 4.4) | 71.9 (± 2.5) |
| RoBERTa-large-FT | B-COPA | 89.0 (± 0.3) | 88.9 (± 2.1) | 89.0 (± 0.8) |
| RoBERTa-large-FT | B-COPA (50%) | 86.1 (± 2.2) | 87.4 (± 1.1) | 85.4 (± 2.9) |
| RoBERTa-large-FT | COPA | 87.7 (± 0.9) | 91.6 (± 1.1) | 85.3 (± 2.0) |

Caption: Table 5: Results of fine-tuned models on Balanced COPA. Easy: instances with superficial cues, Hard: instances without superficial cues.

**Original Claim**

The smaller performance gap between Easy and Hard subsets indicates that training on BCOPA encourages BERT and RoBERTa to rely less on superficial cues.

**Generated Refuted Claim**

The larger performance gap between Easy and Hard subsets indicates that training on BCOPA encourages BERT and RoBERTa to rely more on superficial cues.

Figure 7: The user interface for the refuted claim verification task

stance, "G2S_GAT_BLEU_LDC2015E86" should be "G2S_GIN_BLEU_LDC2015E86". The logical function "and" should be replaced with "or".

**Error Case 4.** It fails to generate a precise program for the approximation word "comparable" (*Ambiguity error*). Currently, the program defines "comparable" as "larger than", which is not accurate enough.

**Error Case 5.** It generates the correct program, but the calculation result is inaccurate due to incorrect float digits in the Python code (*Calculation error*). For instance, Python may output '1.9499999', which is not equal to '1.95'.

# Annotation Post Survey

**Annotator 1:**

- Is the task demonstration clear to you?
  Yes,  clear.
- What do you think is the difficulty of this task?  (1-10 points, 10 points is the most difficult)
  5-6.
- Which part is the most difficult for you? Why?
  Judgment, understanding the way of original author think.
- Do you think the annotation batch is appropriate? What is the maximum batch amount for you in a week?
  Yes. 2 batches in a week during the examination. 4 during vacation.
- Could you provide some advice on how to improve the annotation platform?
  Looping for multiple operations.

---

**Annotator 2:**

- Is the task demonstration clear to you?
  Yes.
- What do you think is the difficulty of this task?  (1-10 points, 10 points is the most difficult)
  6
- Which part is the most difficult for you? Why?
  Table understanding; different parameters in the attributes.
- Do you think the annotation batch is appropriate? What is the maximum batch amount for you in a week?
  Ok. 2-3 batches
- Would you like to attend this session again as a 2-week participation?
  ok.
- Could you provide some advice on how to improve the annotation platform?
  I preferred to write down the annotations on the platform.

---

**Annotator 3:**

- Is the task demonstration clear to you?
  Yes, clear. the difficulty is different between demo and real annotation.
- What do you think is the difficulty of this task?  (1-10 points, 10 points is the most difficult)
  7
- Which part is the most difficult for you? Why?
  Table understanding-vocabulary.
- Do you think the sample amount is appropriate? What is the maximum batch amount for you in a week (1 batch contains 20 samples)?
  10-15   samples for an hour. 50 samples a week.

- Would you like to attend this session again as a 2-week participation?
  Maybe not. But 15 samples offline for a week is ok.

- Could you provide some advice on how to improve the annotation platform?
  I think the current platform is fine for me.

Figure 8: The examples of post-annotation survey questions and the answers of annotators.

# H  Prompts

## H.1  Zero-shot Prompts

```
Table: <input_table>

Claim: <input_claim>

Based on the information in the Table, is the above claim true?
A) the claim is true.
B) the claim is false.
C) it is impossible to tell.
```

## H.2  Few-shot Prompts

```
Read the following table and then answer a question.

Caption: Table 5: Results of fine-tuned models on Balanced COPA. Easy: instances with superficial cues,
Hard: instances without superficial cues.

Table:
|| Model | Training data | Overall | Easy | Hard ||
|| BERT-large-FT | B-COPA | 74.5 (±0.7) | 74.7 (±0.4) | 74.4 (±0.9) ||
|| BERT-large-FT | B-COPA (50%) | 74.3 (±2.2) | 76.8 (±1.9) | 72.8 (±3.1) ||
|| BERT-large-FT | COPA |  76.5 (±2.7) |  83.9 (±4.4) | 71.9 (±2.5) ||
|| RoBERTa-large-FT | B-COPA |  89.0 (±0.3) | 88.9 (±2.1) |  89.0 (±0.8) ||
|| RoBERTa-large-FT | B-COPA (50%) | 86.1 (±2.2) | 87.4 (±1.1) | 85.4 (±2.9) ||
|| RoBERTa-large-FT | COPA | 87.7 (±0.9) |  91.6 (±1.1) | 85.3 (±2.0) ||

Claim: RoBERTa-large outperforms BERT-large when fine-tuned on full and balanced COPA.

Question: Is the above claim true or false? Please directly give the answer.

Answer:
The claim is true.
------
Caption: Table 5: Results of fine-tuned models on Balanced COPA. Easy: instances with superficial cues,
Hard: instances without superficial cues.

Table:
|| Model | Training data | Overall | Easy | Hard ||
|| BERT-large-FT | B-COPA | 74.5 (±0.7) | 74.7 (±0.4) | 74.4 (±0.9) ||
|| BERT-large-FT | B-COPA (50%) | 74.3 (±2.2) | 76.8 (±1.9) | 72.8 (±3.1) ||
|| BERT-large-FT | COPA |  76.5 (±2.7) |  83.9 (±4.4) | 71.9 (±2.5) ||
|| RoBERTa-large-FT | B-COPA |  89.0 (±0.3) | 88.9 (±2.1) |  89.0 (±0.8) ||
|| RoBERTa-large-FT | B-COPA (50%) | 86.1 (±2.2) | 87.4 (±1.1) | 85.4 (±2.9) ||
|| RoBERTa-large-FT | COPA | 87.7 (±0.9) |  91.6 (±1.1) | 85.3 (±2.0) ||

Claim: The difference between RoBERTa-large-FT and BERT-large-FT is 3.8 points on B-COPA,
which is significantly smaller than the difference in COPA.

Question: Is the above claim true or false? Please directly give the answer.

Answer:
The claim is false.
------
Caption:  Table 4: The ablation study on the WoZ2.0 dataset with the joint goal accuracy on the test set.
For``- Hierachical-Attn'', we remove the residual connections between the attention modules in the CMR
decoders and all the attention memory access are based on the output from the LSTM.
For``- MLP'', we further replace the MLP with a single linear layer with the non-linear activation.

Table:
|| Model | Joint Acc. ||
|| COMER | 88.64% ||
|| - Hierachical-Attn | 86.69% ||
|| - MLP | 83.24% ||

Claim: [CONTINUE] The effectiveness of our hierarchical attention design is proved by an accuracy drop
of 1.95% after removing residual connections and the hierarchical stack of our attention modules.

Question: Is the above claim true or false? Please directly give the answer.

Answer:
The claim is true.
------
Caption: Table 4: Scores for different training objectives on the linguistic probing tasks.

Table:
|| Method | Depth| BShift| SubjNum | Tense | CoordInv | Length | ObjNum | TopConst | SOMO | WC ||
|| CMOW-C | 36.2 | 66.0 | 81.1 | 78.7 | 61.7 | 83.9 | 79.1 | 73.6 | 50.4 | 66.8 ||
|| CMOW-R | 35.1 | 70.8 | 82.0 | 80.2 | 61.8 | 82.8 | 79.7 | 74.2 | 50.7 | 72.9 ||
|| CBOW-C | 34.3 | 50.5 | 79.8 | 79.9 | 53.0 | 75.9 | 79.8 | 72.9 | 48.6 | 89.0 ||
|| CBOW-R | 33.0 | 49.6 | 79.3 | 78.4 | 53.6 | 74.5 | 78.6 | 72.0 | 49.6 | 89.5 ||
```

```
Claim: While CMOW-R and CMOW-C perform comparably on most probing tasks,
CMOW-C yields 5 points higher scores on WordContent and BigramShift.

Question:  Is the above claim true or false? Please directly give the answer.

Answer:
The claim is false.

(··· more in-context examples here ···)

------
Caption: <input_caption>

Table: <input_table>

Claim: <input_claim>

Question:  Is the above claim true or false? Please directly give the answer.

Answer:
```

## H.3  Chain-of-Thought Prompts

```
Read the following table and then answer a question.

Caption: Table 5: Results of fine-tuned models on Balanced COPA. Easy: instances with superficial cues,
Hard: instances without superficial cues.

Table:
|| Model | Training data | Overall | Easy | Hard ||
|| BERT-large-FT | B-COPA | 74.5 (±0.7) | 74.7 (±0.4) | 74.4 (±0.9) ||
|| BERT-large-FT | B-COPA (50%) | 74.3 (±2.2) | 76.8 (±1.9) | 72.8 (±3.1) ||
|| BERT-large-FT | COPA |  76.5 (±2.7) |  83.9 (±4.4) | 71.9 (±2.5) ||
|| RoBERTa-large-FT | B-COPA |  89.0 (±0.3) | 88.9 (±2.1) |  89.0 (±0.8) ||
|| RoBERTa-large-FT | B-COPA (50%) | 86.1 (±2.2) | 87.4 (±1.1) | 85.4 (±2.9) ||
|| RoBERTa-large-FT | COPA | 87.7 (±0.9) |  91.6 (±1.1) | 85.3 (±2.0) ||

Claim: RoBERTa-large outperforms BERT-large when fine-tuned on full and balanced COPA.

Question: Is the above claim true or false? Let us think step by step.

Reasoning:
We need to compare the performance of RoBERTa-large and BERT-large on full and balanced COPA.
For full COPA, RoBERTa-large has an overall performance of 87.7 (table[6,2]),
while BERT-large has an overall performance of 76.5 (table[3,2]).
Therefore, RoBERTa-large outperforms BERT-large on full COPA.
For balanced COPA, RoBERTa-large has an overall performance of 89.0 (table[4,2]),
while BERT-large has an overall performance of 74.5 (table[1,2]).
Therefore, RoBERTa-large outperforms BERT-large on balanced COPA.
Finally, since RoBERTa-large outperforms BERT-large on both full and balanced COPA, the Claim is true.

Answer:
Therefore, the claim is true.
------
Caption: Table 5: Results of fine-tuned models on Balanced COPA. Easy: instances with superficial cues,
Hard: instances without superficial cues.

Table:
|| Model | Training data | Overall | Easy | Hard ||
|| BERT-large-FT | B-COPA | 74.5 (±0.7) | 74.7 (±0.4) | 74.4 (±0.9) ||
|| BERT-large-FT | B-COPA (50%) | 74.3 (±2.2) | 76.8 (±1.9) | 72.8 (±3.1) ||
|| BERT-large-FT | COPA |  76.5 (±2.7) |  83.9 (±4.4) | 71.9 (±2.5) ||
|| RoBERTa-large-FT | B-COPA |  89.0 (±0.3) | 88.9 (±2.1) |  89.0 (±0.8) ||
|| RoBERTa-large-FT | B-COPA (50%) | 86.1 (±2.2) | 87.4 (±1.1) | 85.4 (±2.9) ||
|| RoBERTa-large-FT | COPA | 87.7 (±0.9) |  91.6 (±1.1) | 85.3 (±2.0) ||

Claim: The difference between RoBERTa-large-FT and BERT-large-FT is 3.8 points on B-COPA,
which is significantly smaller than the difference in COPA.

Question: Is the above claim true or false? Let us think step by step.

Reasoning:
To verify this claim, we need to compare the performance difference between RoBERTa-large-FT
and BERT-large-FT on B-COPA and COPA.
The difference between RoBERTa-large-FT and BERT-large-FT on B-COPA is 89.0 - 74.5 = 14.5
(table[4,2] - table[1,2]).
The difference between RoBERTa-large-FT and BERT-large-FT on COPA is 87.7 - 76.5 = 11.2
(table[6,2] - table[3,2]).
The difference between RoBERTa-large-FT and BERT-large-FT on B-COPA is significantly larger
than the difference on COPA. Therefore, the Claim is false.

Answer:
Therefore, the claim is false.
------
Caption:  Table 4: The ablation study on the WoZ2.0 dataset with the joint goal accuracy on the test set.
```

```
For``- Hierachical-Attn'', we remove the residual connections between the attention modules in the CMR
decoders and all the attention memory access are based on the output from the LSTM.
For``- MLP'', we further replace the MLP with a single linear layer with the non-linear activation.
Table:
|| Model | Joint Acc. ||
|| COMER | 88.64% ||
|| - Hierachical-Attn | 86.69% ||
|| - MLP | 83.24% ||

Claim: [CONTINUE] The effectiveness of our hierarchical attention design is proved by an accuracy drop
of 1.95% after removing residual connections and the hierarchical stack of our attention modules.

Question: Is the above claim true or false? Let us think step by step.

Reasoning:
To verify this claim, we need to compare the performance of COMER with
and without hierarchical attention design.
The performance of COMER with hierarchical attention design is 88.64% (table[2,2]).
The performance of COMER without hierarchical attention design is 86.69% (table[3,2]).
Therefore, the performance of COMER with hierarchical attention design is higher than
the performance of COMER without hierarchical attention design by 88.64 - 86.69 = 1.95.
Therefore, the Claim is true.

Answer:
Therefore, the claim is true.
------
Caption: Table 4: Scores for different training objectives on the linguistic probing tasks.

Table:
|| Method | Depth| BShift| SubjNum | Tense | CoordInv | Length | ObjNum | TopConst | SOMO | WC ||
|| CMOW-C | 36.2 | 66.0 | 81.1 | 78.7 | 61.7 | 83.9 | 79.1 | 73.6 | 50.4 | 66.8 ||
|| CMOW-R | 35.1 | 70.8 | 82.0 | 80.2 | 61.8 | 82.8 | 79.7 | 74.2 | 50.7 | 72.9 ||
|| CBOW-C | 34.3 | 50.5 | 79.8 | 79.9 | 53.0 | 75.9 | 79.8 | 72.9 | 48.6 | 89.0 ||
|| CBOW-R | 33.0 | 49.6 | 79.3 | 78.4 | 53.6 | 74.5 | 78.6 | 72.0 | 49.6 | 89.5 ||

Claim: While CMOW-R and CMOW-C perform comparably on most probing tasks,
CMOW-C yields 5 points higher scores on WordContent and BigramShift.

Question: Is the above claim true or false? Let us think step by step.

Reasoning:
To verify this claim, we need to compare the performance of CMOW-R
and CMOW-C on WordContent and BigramShift.
The performance of CMOW-R on WordContent is 72.9 (table[2,10]).
The performance of CMOW-C on WordContent is 66.8 (table[1,10]).
The performance of CMOW-R on BigramShift is 70.8 (table[2,2]).
The performance of CMOW-C on BigramShift is 66.0 (table[1,2]).
Therefore, the performance of CMOW-C is higher than
the performance of CMOW-R on WordContent by 66.8 - 72.9 = -6.1.
The performance of CMOW-C is lower than
the performance of CMOW-R on BigramShift by 66.0 - 70.8 = -4.8.
Therefore, the Claim is false.

Answer:
Therefore, the claim is false.

(··· more in-context examples here ···)

------
Caption: <input_caption>

Table: <input_table>

Claim: <input_claim>

Question: Is the above claim true or false? Let us think step by step.

Reasoning:

Answer:
```

## H.4 Program-of-Thoughts Prompts

```
Read the following table and then write Python code to answer a question:
(please call the function equal(a, b) to check whether a and b are equal)

Caption:  Table 4: The ablation study on the WoZ2.0 dataset with the joint goal accuracy on the test set.
For``- Hierachical-Attn'', we remove the residual connections between the attention modules in the CMR
decoders and all the attention memory access are based on the output from the LSTM.
For``- MLP'', we further replace the MLP with a single linear layer with the non-linear activation.

Table:
|| Model | Joint Acc. ||
|| COMER | 88.64% ||
|| - Hierachical-Attn | 86.69% ||
```

```
|| - MLP | 83.24% ||
```

**Claim**: [CONTINUE] The effectiveness of our hierarchical attention design is proved by
an accuracy drop of 1.95% after removing residual connections
and the hierarchical stack of our attention modules.

**Question**: Based on the information in the table, is the above claim true or false?

```python
# Python Code, return ans
COMER_acc = 88.64
COMER_acc_no_residual = 86.69
accuracy_drop = COMER_acc - COMER_acc_no_residual
ans = equal(accuracy_drop, 1.95)
```
------
Read the following table and then write Python code to answer a question:
(please call the function equal(a, b) to check whether a and b are equal)

**Caption**: Table 3: Ablation study of capsule net and word-level attention on Wikidata dataset.

**Table**:
```
|| Recall | 0.1 | 0.2 | 0.3 | AUC ||
|| -Word-ATT | 0.648 | 0.515 | 0.395 | 0.389 ||
|| -Capsule | 0.635 | 0.507 | 0.413 | 0.386 ||
|| Our Model | 0.650 | 0.519 | 0.422 | 0.405 ||
```

**Claim**: According to the table, the drop of precision demonstrates
that the word-level attention is quite useful.

**Question**: Based on the information in the table, is the above claim true or false?

```python
# Python Code, return ans
our_model_recalls = [0.650, 0.519, 0.422, 0.405]
without_word_att_recalls = [0.648, 0.515, 0.395, 0.389]
ans = True
for i in range(4):
    if our_model_recalls[i] < without_word_att_recalls[i]:
        ans = False
        break
```
------
Read the following table and then write Python code to answer a question:
(please call the function equal(a, b) to check whether a and b are equal)

**Caption**: Table 4: Scores for different training objectives on the linguistic probing tasks.

**Table**:
```
|| Method | Depth| BShift| SubjNum | Tense | CoordInv | Length | ObjNum | TopConst | SOMO | WC ||
|| CMOW-C | 36.2 | 66.0 | 81.1 | 78.7 | 61.7 | 83.9 | 79.1 | 73.6 | 50.4 | 66.8 ||
|| CMOW-R | 35.1 | 70.8 | 82.0 | 80.2 | 61.8 | 82.8 | 79.7 | 74.2 | 50.7 | 72.9 ||
|| CBOW-C | 34.3 | 50.5 | 79.8 | 79.9 | 53.0 | 75.9 | 79.8 | 72.9 | 48.6 | 89.0 ||
|| CBOW-R | 33.0 | 49.6 | 79.3 | 78.4 | 53.6 | 74.5 | 78.6 | 72.0 | 49.6 | 89.5 ||
```

**Claim**: While CMOW-R and CMOW-C perform comparably on most probing tasks,
CMOW-C yields 5 points higher scores on WordContent and BigramShift.

**Question**: Based on the information in the table, is the above claim true or false?

```python
# Python Code, return ans
CMOW_C_score_on_WC = 66.8
CMOW_C_score_on_BShift = 66.0
CMOW_R_score_on_WC = 72.9
CMOW_R_score_on_BShift = 70.8
ans = equal(CMOW_C_score_on_WC - CMOW_R_score_on_WC, 5)
and equal(CMOW_C_score_on_BShift - CMOW_R_score_on_BShift, 5)
```

(··· **more in-context examples here** ···)

------
Read the following table and then write Python code to answer a question:
(please call the function equal(a, b) to check whether a and b are equal)

**Caption**: <input_caption>

**Table**: <input_table>

**Claim**: <input_claim>

**Question**: Based on the information in the table, is the above claim true or false?

```python
# Python Code, return ans
```

**A. The calculation result is wrong.**

| Model | T | #P | B | C |
|---|---|---|---|---|
| Seq2SeqB (Beck et al., 2018) | S | 28,4M | 21.7 | 49.1 |
| GGNN2Seq (Beck et al., 2018) | S | 28.3M | 23.3 | 50.4 |
| Seq2SeqB (Beck et al., 2018) | E | 142M | 26.6 | 52.5 |
| GGNN2Seq (Beck et al., 2018) | E | 141M | 27.5 | 53.5 |
| DCGCN (ours) | S | 19.1M | 27.9 | 57.3 |
| | E | 92.5M | **30.4** | **59.6** |

Table 2: Main results on AMR17. GCNSEQ (Damonte and Cohen, 2019) achieves 24.5 BLEU points. #P shows the model size in terms of parameters; "S" and "E" denote single and ensemble models, respectively.

**Claim:** For example, the single DCGCN model gains 5.9 more BLEU points than the single models of Seq2SeqB on AMR17.

---

**B. The approximation word is wrong.**

| GCN | B | C | GCN | B | C |
|---|---|---|---|---|---|
| +RC (2) | 16.8 | 48.1 | +RC+LA (2) | 18.3 | 47.9 |
| +RC (4) | 18.4 | 49.6 | +RC+LA (4) | 18.0 | 51.1 |
| +RC (6) | 19.9 | 49.7 | +RC+LA (6) | 21.3 | 50.8 |
| +RC (9) | **21.1** | 50.5 | +RC+LA (9) | **22.0** | 52.6 |
| +RC (10) | 20.7 | **50.7** | +RC+LA (10) | 21.2 | **52.9** |
| DCGCN1 (9) | 22.9 | 53.0 | DCGCN3 (27) | 24.8 | 54.7 |
| DCGCN2 (18) | 24.2 | 54.4 | DCGCN4 (36) | **25.5** | **55.4** |

Table 6: Comparisons with baselines. +RC denotes GCNs with residual connections. +RC+LA refers to GCNs with both residual connections and layer aggregations. DCGCN$i$ represents our model with $i$ blocks, containing $i \times (n + m)$ layers. The number of layers for each model is shown in parenthesis.

**Claim:** However, when we set the number to 18, GCN+RC+LA achieves a BLEU score of 19.4, which is significantly worse than the BLEU score obtained by DCGCN2 (23.3).

---

**C. The claim is partially right.**

Table 4: Scores for different training objectives on the linguistic probing tasks.

| Method | Depth | BShift | SubjNum | Tense | CoordInv | Length | ObjNum | TopConst | SOMO | WC |
|---|---|---|---|---|---|---|---|---|---|---|
| CMOW-C | **36.2** | 66.0 | 81.1 | 78.7 | 61.7 | **83.9** | 79.1 | 73.6 | 50.4 | 66.8 |
| CMOW-R | 35.1 | **70.8** | **82.0** | **80.2** | **61.8** | 82.8 | **79.7** | **74.2** | **50.7** | **72.9** |
| CBOW-C | **34.3** | **50.5** | **79.8** | **79.9** | 53.0 | **75.9** | **79.8** | **72.9** | 48.6 | 89.0 |
| CBOW-R | 33.0 | 49.6 | 79.3 | 78.4 | **53.6** | 74.5 | 78.6 | 72.0 | **49.6** | **89.5** |

**Claim:** While CMOW-R and CMOW-C perform comparably on most probing tasks, CMOW-C yields 5 points lower scores on WordContent and BigramShift.
*BShift column doesn't satisfy the claim.*

Figure 9: The refuted claims cases *A* to *C*. Case *A* represents the calculation result is wrong. Case *B* represents the approximation word is wrong. Case *C* represents the claim is partially right.

**D. The values in the claim do not match.**

| Model | T | #P | B | C |
|---|---|---|---|---|
| Seq2SeqB (Beck et al., 2018) | S | 28,4M | 21.7 | 49.1 |
| GGNN2Seq (Beck et al., 2018) | S | 28.3M | 23.3 | 50.4 |
| Seq2SeqB (Beck et al., 2018) | E | 142M | 26.6 | 52.5 |
| GGNN2Seq (Beck et al., 2018) | E | 141M | 27.5 | 53.5 |
| DCGCN (ours) | S | **19.1M** | 27.9 | 57.3 |
| | E | 92.5M | **30.4** | **59.6** |

Table 2: Main results on AMR17. GCNSEQ (Damonte and Cohen, 2019) achieves 24.5 BLEU points. #P shows the model size in terms of parameters; "S" and "E" denote single and ensemble models, respectively.

**Claim:** Our single model achieves 27.6 BLEU points, which is the new state-of-the-art result for single models.

---

**E. The operation type is wrong.**

| GCN | B | C | GCN | B | C |
|---|---|---|---|---|---|
| +RC (2) | 16.8 | 48.1 | +RC+LA (2) | 18.3 | 47.9 |
| +RC (4) | 18.4 | 49.6 | +RC+LA (4) | 18.0 | 51.1 |
| +RC (6) | 19.9 | 49.7 | +RC+LA (6) | 21.3 | 50.8 |
| +RC (9) | **21.1** | 50.5 | +RC+LA (9) | **22.0** | 52.6 |
| +RC (10) | 20.7 | **50.7** | +RC+LA (10) | 21.2 | **52.9** |
| DCGCN1 (9) | 22.9 | 53.0 | DCGCN3 (27) | 24.8 | 54.7 |
| DCGCN2 (18) | 24.2 | 54.4 | DCGCN4 (36) | **25.5** | **55.4** |

Table 6: Comparisons with baselines. +RC denotes GCNs with residual connections. +RC+LA refers to GCNs with both residual connections and layer aggregations. DCGCN$i$ represents our model with $i$ blocks, containing $i \times (n + m)$ layers. The number of layers for each model is shown in parenthesis.

**Claim:** GCN+RC+LA (9) is better than DCGCN1.

---

Figure 10: The refuted claims cases *D* and *E*. Case *D* represents the values in the claim do not match. Case *E* represents the operation type is wrong.

**Error case 1 (43.9%)** : Gold Label: **Supported**  Prediction Label:  **NEI**

Table 3: Precision, recall and F-measure for methods using the top 1,000 words with the highest number of contexts.

|  | Lang | Corpus | Patt | DSim | SLQS | TF | DF | DocSub | HClust |
|---|---|---|---|---|---|---|---|---|---|
| $\mathcal{P}$ | EN | Europarl | **0.1173** | 0.0366 | 0.0503 | 0.0554 | 0.0548 | 0.0443 | 0.0761 |
| | | Ted Talks | **0.1125** | 0.0301 | 0.0382 | 0.0425 | 0.0441 | 0.0710 | 0.0664 |
| | PT | Europarl | 0.5163 | 0.3330 | 0.5257 | 0.6109 | 0.5984 | **0.7311** | 0.5676 |
| | | Ted Talks | 0.5387 | 0.2907 | 0.5300 | 0.6117 | 0.6159 | **0.6533** | 0.5656 |
| $\mathcal{R}$ | EN | Europarl | 0.0396 | 0.3999 | 0.5499 | **0.6045** | 0.5887 | 0.0023 | 0.0017 |
| | | Ted Talks | 0.0018 | 0.4442 | 0.5377 | 0.5657 | **0.6077** | 0.2666 | 0.0019 |
| | PT | Europarl | 0.0111 | 0.3554 | 0.5795 | **0.6727** | 0.5184 | 0.0053 | 0.0012 |
| | | Ted Talks | 0.0004 | 0.3142 | 0.5484 | **0.6877** | 0.5515 | 0.4706 | 0.0011 |
| $\mathcal{F}$ | EN | Europarl | 0.0591 | 0.0671 | 0.0922 | **0.1015** | 0.1003 | 0.0044 | 0.0033 |
| | | Ted Talks | 0.0035 | 0.0564 | 0.0713 | 0.0791 | 0.0822 | **0.1121** | 0.0037 |
| | PT | Europarl | 0.0217 | 0.3438 | 0.5513 | **0.6403** | 0.5555 | 0.0105 | 0.0024 |
| | | Ted Talks | 0.0008 | 0.3020 | 0.5390 | **0.6475** | 0.5819 | 0.5471 | 0.0022 |

**Claim**: Analyzing Table 3, we can observe that all values of precision using the Portuguese corpora have higher scores when compared with the English corpora.

---

**Error case 2 (38.7%):** Gold Label:  **Refuted**  Prediction Label: **NEI**

Table 3: Precision, recall and F-measure for methods using the top 1,000 words with the highest number of contexts.

|  | Lang | Corpus | Patt | DSim | SLQS | TF | DF | DocSub | HClust |
|---|---|---|---|---|---|---|---|---|---|
| $\mathcal{P}$ | EN | Europarl | **0.1173** | 0.0366 | 0.0503 | 0.0554 | 0.0548 | 0.0443 | 0.0761 |
| | | Ted Talks | **0.1125** | 0.0301 | 0.0382 | 0.0425 | 0.0441 | 0.0710 | 0.0664 |
| | PT | Europarl | 0.5163 | 0.3330 | 0.5257 | 0.6109 | 0.5984 | **0.7311** | 0.5676 |
| | | Ted Talks | 0.5387 | 0.2907 | 0.5300 | 0.6117 | 0.6159 | **0.6533** | 0.5656 |
| $\mathcal{R}$ | EN | Europarl | 0.0396 | 0.3999 | 0.5499 | **0.6045** | 0.5887 | 0.0023 | 0.0017 |
| | | Ted Talks | 0.0018 | 0.4442 | 0.5377 | 0.5657 | **0.6077** | 0.2666 | 0.0019 |
| | PT | Europarl | 0.0111 | 0.3554 | 0.5795 | **0.6727** | 0.5184 | 0.0053 | 0.0012 |
| | | Ted Talks | 0.0004 | 0.3142 | 0.5484 | **0.6877** | 0.5515 | 0.4706 | 0.0011 |
| $\mathcal{F}$ | EN | Europarl | 0.0591 | 0.0671 | 0.0922 | **0.1015** | 0.1003 | 0.0044 | 0.0033 |
| | | Ted Talks | 0.0035 | 0.0564 | 0.0713 | 0.0791 | 0.0822 | **0.1121** | 0.0037 |
| | PT | Europarl | 0.0217 | 0.3438 | 0.5513 | **0.6403** | 0.5555 | 0.0105 | 0.0024 |
| | | Ted Talks | 0.0008 | 0.3020 | 0.5390 | **0.6475** | 0.5819 | 0.5471 | 0.0022 |

**Claim**: Analyzing Table 3, we can observe that all values of precision using the English corpora have higher scores when compared with the Portuguese corpora.

---

**Error case 3 (7.6%)** : Gold Label:  **Refuted**  Prediction Label:  **Supported**

| Model | B | C |
|---|---|---|
| DCGCN4 | 25.5 | 55.4 |
| Encoder Modules | | |
| -Linear Combination | 23.7 | 53.2 |
| -Global Node | 24.2 | 54.6 |
| -Direction Aggregation | 24.6 | 54.6 |
| -Graph Attention | 24.9 | 54.7 |
| -Global Node&Linear Combination | 22.9 | 52.4 |
| Decoder Modules | | |
| -Coverage Mechanism | 23.8 | 53.0 |

Table 9: Ablation study for modules used in the graph encoder and the LSTM decoder

**Claim:** With the coverage mechanism, the result drops by 1.7/2.4 points for B/C scores.

Figure 11: Error Cases 1-3 for InstructGPT in the zero-shot setting.

Error case 4 **(4.6%) :** Gold Label: **NEI**  Prediction Label:  **Supported**

|  | in-domain SQuAD | | out-of-domain QA-SRL | |
|---|---|---|---|---|
|  | EM | F1 | EM | F1 |
| MQAN | 31.76 | 75.37 | **10.99** | 50.10 |
| +coverage | **32.67** | **76.83** | 10.63 | **50.89** |
| BIDAF (ELMO) | 70.43 | 79.76 | 28.35 | 49.98 |
| +coverage | **71.07** | **80.15** | **30.58** | **52.43** |

Table 3: Impact of using coverage for improving generalization across the datasets of similar tasks. Both models are trained on the SQuAD training data.

**Claim:** This is particularly true for the BIDAF model.

---

Error case 5 **(2.8%) :** Gold Label: **NEI**  Prediction Label:  **Refuted**

TABLE V

WORD INTRUSION TEST RESULTS: CORRECT ANSWERS OUT OF 300 QUESTIONS

|  | GloVe | Imparted |
|---|---|---|
| Participants 1 to 5 | 80/88/82/78/97 | 212/170/207/229/242 |
| Mean/Std | 85/6.9 | 212/24.4 |

**Claim:** These results show that the questionnaire takers had an average accuracy of 98.2% in answering word intrusion questions for words associated with meanings imparted by standard word embeddings

---

| GCN | B | C | GCN | B | C |
|---|---|---|---|---|---|
| +RC (2) | 16.8 | 48.1 | +RC+LA (2) | 18.3 | 47.9 |
| +RC (4) | 18.4 | 49.6 | +RC+LA (4) | 18.0 | 51.1 |
| +RC (6) | 19.9 | 49.7 | +RC+LA (6) | 21.3 | 50.8 |
| +RC (9) | **21.1** | 50.5 | +RC+LA (9) | **22.0** | 52.6 |
| +RC (10) | 20.7 | **50.7** | +RC+LA (10) | 21.2 | **52.9** |
| DCGCN1 (9) | 22.9 | 53.0 | DCGCN3 (27) | 24.8 | 54.7 |
| DCGCN2 (18) | 24.2 | 54.4 | DCGCN4 (36) | **25.5** | **55.4** |

Table 6: Comparisons with baselines. +RC denotes GCNs with residual connections. +RC+LA refers to GCNs with both residual connections and layer aggregations. DCGCN$i$ represents our model with $i$ blocks, containing $i \times (n + m)$ layers. The number of layers for each model is shown in parenthesis.

**Claim:** For example, GCN+RC+LA (10) achieves a BLEU score of 21.2, which is worse than GCN+RC+LA (9).

Figure 12: Error Cases 4-6 for InstructGPT in the zero-shot setting.

## Error case 1

| Dataset | Metric | Illinois | IlliCons | Rahman and Ng (2012) | KnowFeat | KnowCons | KnowComb |
|---------|--------|----------|----------|----------------------|----------|----------|----------|
| *Winograd* | Precision | 51.48 | 53.26 | 73.05 | 71.81 | 74.93 | **76.41** |
| *WinoCoref* | AntePre | 68.37 | 74.32 | — | 88.48 | 88.95 | **89.32** |

Table 7: Performance results on *Winograd* and *WinoCoref* datasets. All our three systems are trained on *WinoCoref*, and we evaluate the predictions on both datasets. Our systems improve over the baselines by over than 20% on *Winograd* and over 15% on *WinoCoref*.

**Claim**: The best performing system is KnowComb. It improves by over 20% over a state-of-art general coreference system on Winograd. On the WinoCoref dataset, it improves by 15%.

**gold**: supports, **prediction**: refutes,

**programs**: ["winograd_baseline = 73.05\n winograd_knowcomb = 76.41\n winocoref_baseline = 88.48\nwinocoref_knowcomb = 89.32\nans = (winograd_knowcomb - winograd_baseline > 20) and (winocoref_knowcomb - winocoref_baseline > 15)"

---

## Error case 2

TABLE I
WERs (%) AND DCE OF DIFFERENT SPEECH ENHANCEMENT METHODS ON LIBRISPEECH + DEMAND TEST SET

| Method | WER (%) | DCE |
|--------|---------|-----|
| No enhancement | 17.3 | 0.828 |
| Wiener filter | 19.5 | 0.722 |
| Minimizing DCE | 15.8 | **0.269** |
| FSEGAN | 14.9 | 0.291 |
| AAS ($w_{AC} = 1, w_{AD} = 0$) | 15.6 | 0.330 |
| AAS ($w_{AC} = 1, w_{AD} = 10^5$) | **14.4** | 0.303 |
| Clean speech | 5.7 | 0.0 |

**Claim**: The AAS method with weights wAC=1 and wAD=105 shows the lowest WER and DCE.

**gold**: refutes,

**prediction**: supports,

**programs**: [
AAS_wAC_1_wAD_0_WER = 15.6\n
AAS_wAC_1_wAD_105_WER = 14.4\n
AAS_wAC_1_wAD_0_DCE = 0.330\n
AAS_wAC_1_wAD_105_DCE = 0.303\n
ans = (AAS_wAC_1_wAD_105_WER < AAS_wAC_1_wAD_0_WER)
and (AAS_wAC_1_wAD_105_DCE < AAS_wAC_1_wAD_0_DCE)

---

## Error case 3

| Model | BLEU | METEOR |
|-------|------|--------|
| LDC2015E86 | | |
| Konstas et al. (2017) | 22.00 | - |
| Song et al. (2018) | 23.28 | 30.10 |
| Cao et al. (2019) | 23.50 | - |
| Damonte et al.(2019) | 24.40 | 23.60 |
| Guo et al. (2019) | **25.70** | - |
| S2S | $22.55 \pm 0.17$ | $29.90 \pm 0.31$ |
| G2S-GIN | $22.93 \pm 0.20$ | $29.72 \pm 0.09$ |
| G2S-GAT | $23.42 \pm 0.16$ | $29.87 \pm 0.14$ |
| G2S-GGNN | $24.32 \pm 0.16$ | $\mathbf{30.53} \pm 0.30$ |
| LDC2017T10 | | |
| Back et al. (2018) | 23.30 | - |
| Song et al. (2018) | 24.86 | 31.56 |
| Damonte et al.(2019) | 24.54 | 24.07 |
| Cao et al. (2019) | 26.80 | - |
| Guo et al. (2019) | 27.60 | - |
| S2S | $22.73 \pm 0.18$ | $30.15 \pm 0.14$ |
| G2S-GIN | $26.90 \pm 0.19$ | $32.62 \pm 0.04$ |
| G2S-GAT | $26.72 \pm 0.20$ | $32.52 \pm 0.02$ |
| G2S-GGNN | $\mathbf{27.87} \pm 0.15$ | $\mathbf{33.21} \pm 0.15$ |

Table 2: BLEU and METEOR scores on the test set of LDC2015E86 and LDC2017T10 datasets.

**gold**: supports, **prediction**: refutes

**programs** : [
"G2S_GIN_BLEU_LDC2015E86 = 22.93\n
G2S_GAT_BLEU_LDC2015E86 = 23.42\n
G2S_GGNN_BLEU_LDC2015E86 = 24.32\n
G2S_GIN_METEOR_LDC2015E86 = 29.72\n
G2S_GAT_METEOR_LDC2015E86 = 29.87\n
G2S_GGNN_METEOR_LDC2015E86 = 30.53\n
G2S_GIN_BLEU_LDC2017T10 = 26.90\n
G2S_GAT_BLEU_LDC2017T10 = 26.72\n
G2S_GGNN_BLEU_LDC2017T10 = 27.87\n
G2S_GIN_METEOR_LDC2017T10 = 32.62\n
G2S_GAT_METEOR_LDC2017T10 = 32.52\n
G2S_GGNN_METEOR_LDC2017T10 = 33.21\n
ans = (G2S_GAT_BLEU_LDC2015E86 > G2S_GIN_BLEU_LDC2015E86)
and (G2S_GGNN_BLEU_LDC2015E86 > G2S_GAT_BLEU_LDC2015E86)
and (G2S_GAT_METEOR_LDC2015E86 > G2S_GIN_METEOR_LDC2015E86)
and (G2S_GGNN_METEOR_LDC2015E86 > G2S_GAT_METEOR_LDC2015E86)
and (G2S_GAT_BLEU_LDC2017T10 > G2S_GIN_BLEU_LDC2017T10)
and (G2S_GGNN_BLEU_LDC2017T10 > G2S_GAT_BLEU_LDC2017T10)
and (G2S_GAT_METEOR_LDC2017T10 > G2S_GIN_METEOR_LDC2017T10) and (G2S_GGNN_METEOR_LDC2017T10 > G"

**claim**: This suggests that graph encoders based on gating mechanisms are very effective in text generation models.

Figure 13: Error Cases 1-3 for Program-of-Thoughts. Error Case 1 exhibits incorrect entity linking (*Grounding error*) and incorrect operation (*Program error*). Error Case 2 exhibits incomplete entity linking (*Grounding error*). Error Case 3 exhibits *Program error* since it fails to generate a correct program.

**Error case 4**

| System | ROUGE-1 | | | ROUGE-2 | | | Sentence-Level | | |
|---|---|---|---|---|---|---|---|---|---|
| | R (%) | P (%) | F (%) | R (%) | P (%) | F (%) | R (%) | P (%) | F (%) |
| **ILP** | 24.5 | 41.1 | 29.3±0.5 | 7.9 | 15.0 | 9.9±0.5 | 13.6 | 22.6 | 15.6±0.4 |
| **Sum-Basic** | 28.4 | 44.4 | 33.1±0.5 | 8.5 | 15.6 | 10.4±0.4 | 14.7 | 22.9 | 16.7±0.5 |
| **KL-Sum** | 39.5 | 34.6 | 35.5±0.5 | 13.0 | 12.7 | 12.3±0.5 | 15.2 | 21.1 | 16.3±0.5 |
| **LexRank** | 42.1 | 39.5 | 38.7±0.5 | 14.7 | 15.3 | 14.2±0.5 | 14.3 | 21.5 | 16.0±0.5 |
| **MEAD** | 45.5 | 36.5 | 38.5±0.5 | 17.9 | 14.9 | 15.4±0.5 | 27.8 | 29.2 | 26.8±0.5 |
| **SVM** | 19.0 | 48.8 | 24.7±0.8 | 7.5 | 21.1 | 10.0±0.5 | 32.7 | 34.3 | 31.4±0.4 |
| **LogReg** | 26.9 | 34.5 | 28.7±0.6 | 6.4 | 9.9 | 7.3±0.4 | 12.2 | 14.9 | 12.7±0.5 |
| **LogReg$^r$** | 28.0 | 34.8 | 29.4±0.6 | 6.9 | 10.4 | 7.8±0.4 | 12.1 | 14.5 | 12.5±0.5 |
| **HAN** | 31.0 | 42.8 | 33.7±0.7 | 11.2 | 17.8 | 12.7±0.5 | 26.9 | 34.1 | 32.4±0.5 |
| **HAN+pretrainT** | 32.2 | 42.4 | 34.4±0.7 | 11.5 | 17.5 | 12.9±0.5 | 29.6 | 35.8 | 32.2±0.5 |
| **HAN+pretrainU** | 32.1 | 42.1 | 33.8±0.7 | 11.6 | 17.6 | 12.9±0.5 | 30.1 | 35.6 | 32.3±0.5 |
| **HAN$^r$** | 38.1 | 40.5 | **37.8**±0.5 | 14.0 | 17.1 | **14.7**±0.5 | 32.5 | 34.4 | **33.4**±0.5 |
| **HAN+pretrainT$^r$** | 37.9 | 40.4 | **37.6**±0.5 | 13.5 | 16.8 | **14.4**±0.5 | 32.5 | 34.4 | **33.4**±0.5 |
| **HAN+pretrainU$^r$** | 37.9 | 40.4 | **37.6**±0.5 | 13.6 | 16.9 | **14.4**±0.5 | 33.9 | 33.8 | **33.8**±0.5 |

Table 1: Results of thread summarization. 'HAN' models are our proposed approaches adapted from the hierarchical attention networks (Yang et al. 2016). The models can be pretrained using unlabeled threads from TripAdvisor ('T') and Ubuntuforum ('U'). $r$ indicates a redundancy removal step is applied. We report the variance of F-scores across all threads ('±'). A redundancy removal step improves recall scores (shown in gray) of the HAN models and boosts performance.

**claim**: Pretraining the HAN models, although intuitively promising, yields only comparable results with those without.
**gold**: supports, **prediction**: refutes
**Programs**:
["HAN_F_score = 37.8\n
HAN_pretrainT_F_score = 37.6\n
HAN_pretrainU_F_score = 37.6\n
ans = (HAN_F_score - HAN_pretrainT_F_score == 0.2) and
(HAN_F_score - HAN_pretrainU_F_score == 0.2)"]

---

**Error case 5**

| Model | Joint Acc. |
|---|---|
| COMER | 88.64% |
| - Hierachical-Attn | 86.69% |
| - MLP | 83.24% |

Table 4: The ablation study on the WoZ2.0 dataset with the joint goal accuracy on the test set. For "-Hierachical-Attn", we remove the residual connections between the attention modules in the CMR decoders and all the attention memory access are based on the output from the LSTM. For "- MLP", we further replace the MLP with a single linear layer with the non-linear activation.

**Claim**: The effectiveness of our hierarchical attention design is proved by an accuracy drop of 1.95% after removing residual connections and the hierarchical stack of our attention modules.",
**gold**: supports,
**prediction**: refutes,
**programs**: ["
COMER_acc = 88.64\n
COMER_acc_no_residual = 86.69\n
accuracy_drop = COMER_acc -
COMER_acc_no_residual\n
ans = (accuracy_drop == 1.95)"]

Figure 14: Error Cases 4 and 5 for Program-of-Thoughts. Error Case 4 exhibits *Ambiguity error* since it fails to generate a precise program for the approximation word "comparable". Error Case 5 exhibits *Calculation error* since it generates the correct program, but the calculation result is inaccurate due to incorrect float digits in the Python code.