# OpenReview forum: "SCITAB: A Challenging Benchmark for Compositional Reasoning and Claim Verification on Scientific Tables"
_EMNLP/2023/Conference — EMNLP 2023 Main_

### Official Review · Reviewer_BWAe · 2023-07-31

**Typos Grammar Style And Presentation Improvements:** 1. I suggest using different terms be…
**Soundness:** 3

**Excitement:**

4: Strong: This paper deepens the understanding of some phenomenon or lowers the barriers to an existing research direction.

**Missing References:**

Not that I know of.

**Paper Topic And Main Contributions:**

This paper introduces a new dataset called SCITAB, which is designed to address the limitations of existing benchmarks for scientific fact-checking. The SCITAB dataset consists of 1225 challenging scientific claims that require compositional reasoning with scientific tables. These claims are derived from actual scientific statements, and the evidence is presented in the form of tables, which mirrors real-world fact-checking scenarios. The authors have established benchmarks on SCITAB using state-of-the-art models and have identified unique challenges, such as ambiguous expressions and irrelevant claims. The main contribution of this paper is the creation of the SCITAB dataset, which provides a new resource for the research community to further explore and improve scientific fact-checking. The authors have made the dataset publicly available for further research.

**Questions For The Authors:**

QA: I'm not very clear about the method for generating unverifiable claims described in Sec 2.2. Why is this method ("... generate candidate NEI claims by prompting the model with the original table and the instruction ...") able to generate unverifiable claims? Why not remove/replace parts of the table for this purpose? Although manual verification is appreciated, I feel like it's never a bad thing to save the use of human labor for other purposes (like scaling up the datasets).

QB: As in the paper, most reasoning functions in table reasoning are simple, like add, subtraction, lookup, etc. However, when these functions are used together, the results of LLMs turn out to be poor. I would like to see some analysis of the compositionality problem in table-based reasoning using this dataset.

QC: Have you tried other advanced prompting techniques for reasoning, such as self-consistency, least-to-most prompting, etc.?

QD: Can the human-machine collaboration pipeline be optimized? Are there any guidelines for deciding which parts of the data construction process should use human labor?

QE: Given the fact that 20% of the errors are caused by miscalculations, I wonder if equipping the models with external tools (like calculators) will help. Parsing code from the text for the Python interpreter may be too heavy for simple calculations.


**Reasons To Accept:**

I think the major reason to accept this paper is that SCITAB is a manually verified, high-quality dataset, presenting a crucial challenge for the research community, particularly considering that LLMs have achieved exceptional results on many traditional tasks. Besides, the detailed annotation and manual filtration steps implemented in the dataset ensure the high quality of SCITAB. To understand why these models fail, this paper also provides an interesting error analysis that offers an understanding of why LLMs fail to accomplish this task.

**Reasons To Reject:**

1. The human-machine collaboration for data annotation (i.e., machines generate data, humans verify data) is widely adopted in this work. Although it is commendable, I'd like to see some analysis of the best practice of using "precious" human labor in the construction of a dataset.
2. Certain manual annotation experiments are conducted by only one annotator (Line 122), which could result in undetectable biases.
3. I would like to see some in-depth evaluation and analysis for LLMs with SCITAB in the main body of the paper, for example, the compositionality in reasoning (see QC). Perhaps more experiments or case studies should be incorporated into the main content. Some material from the appendix could potentially be included to strengthen the argument that SCITAB is a challenging dataset.

**Reproducibility:**

4: Could mostly reproduce the results, but there may be some variation because of sample variance or minor variations in their interpretation of the protocol or method.

**Reviewer Confidence:**

4: Quite sure. I tried to check the important points carefully. It's unlikely, though conceivable, that I missed something that should affect my ratings.

---

> ### Author Rebuttal · Authors · 2023-08-28
>
> We thank Reviewer BWAe for the thoughtful feedback. We are encouraged by the positive remarks on our work, especially highlighting the introduction of the SCITAB dataset, the high quality achieved through manual verification, presenting a crucial challenge for the research community, and the insightful error analysis which offers an understanding of why LLMs fail on this task. We appreciate the valuable insights behind the comments and also want to clarify a few points.
>
>
> > *A. The human-machine collaboration for data annotation (i.e., machines generate data, humans verify data) is widely adopted in this work. Although it is commendable, I'd like to see some analysis of the best practice of using "precious" human labor in the construction of a dataset.*
>
> Our final data creation pipeline undergoes repetitive testing and revision until it reaches its current form. In our pilot annotation, we found that manual verification played the most essential role in the validation of claims marked as "Not Enough Information" (NEI). Initially, we planned to rely solely on LLMs for generating NEI claims. Our criteria for the NEI claim is that "the claim should be fluent, logical, and relevant to the table. However, the claim cannot be verified as true or false solely based on the information in the table." However, after a careful examination of the LLM output, we found that LLM tends to generate claims that are either not logical or irrelevant to the table content. Therefore, human efforts are required to further select NEI claims that meet our criteria. Out of 900 initial NEI claims generated by LLMs, manual verification narrowed them down to only 355 claims, taking up 40% of the original count. While it may not have served a crucial role in filtering NEI claims, human verification also safeguarded the data quality in other annotation processes. For example, among the "supported" claims originally appearing in the scientific paper, human validation still identified 10 examples that were actually not supported  (e.g., wrong number matching, etc.). Following the reviewer's suggestion, we will add the above discussion on the level of human contributions to each part of our data creation pipeline to the camera-ready version.
>
>
> > *B. Certain manual annotation experiments are conducted by only one annotator (Line 122), which could result in undetectable biases.*
>
> First, we want to clarify that the only manual annotation experiment with one annotator is the initial claim extraction phase. We assign a single annotator for this phrase because identifying whether a sentence is a scientific claim is a relatively easy and objective task. During our pilot annotation, we found that a single trained annotator can achieve over 95% accuracy in filtering scientific claims. Therefore, to save the cost of human labor, we use a single annotator. To further safeguard the quality, during the subsequent claim verification process, we include an option to mark the claim as "Discard: It's not a claim, or it's an incomplete, or not grammatically correct sentence." We will include the above annotation details in Section 2.1 - Data Preparation.
>
>
>
> > *C. I would like to see some in-depth evaluation and analysis for LLMs with SCITAB in the main body of the paper, for example, the compositionality in reasoning (see QB). Perhaps more experiments or case studies should be incorporated into the main content. Some material from the appendix could potentially be included to strengthen the argument that SCITAB is a challenging dataset.*
>
> We agree that in-depth evaluation and analysis of LLMs are both necessary and interesting. To clarify, we have indeed conducted in-depth evaluations and analyses of LLMs in several sections of our paper. These include 1) Section 4.2 Error Analysis, which investigates why LLMs (InstructGPT, GPT-4, and PoT) fail with the SCITAB dataset; 2) Appendix F Error Cases for InstructGPT; and 3) Appendix G Error Cases for PoT, showcasing specific error cases along with their explanations. However, due to space limits, many of these explanations are given in the appendix, such as Lines 924 - 950 and Lines 952 - 983. Following the reviewer's suggestion, we will include some of these details in the main body of the camera-ready version.
>
>
> > *QA: I'm not very clear about the method for generating unverifiable claims described in Sec 2.2. Why is this method ("... generate candidate NEI claims by prompting the model with the original table and the instruction ...") able to generate unverifiable claims? Why not remove/replace parts of the table for this purpose? Although manual verification is appreciated, I feel like it's never a bad thing to save the use of human labor for other purposes (like scaling up the datasets).*
>
> First, regarding why this method generates unverifiable claims, we have explained the reason in the paper from Lines 163 to 167: "However, as LLMs tend to generate content that might not always be grounded in the provided data, many of the generated claims turn out to be relevant but unverifiable with respect to the table." The reason behind this phenomenon is LLMs often generate outputs that are not grounded in the input data, a phenomenon known as "hallucination" [1]. In the NEI claim generation, the hallucination behavior of LLMs results in "many of the generated claims turn out to be relevant but unverifiable with respect to the table."
>
> Second, regarding why not remove/replace parts of the table, our primary goal is to produce a realistic scientific fact-checking dataset, which necessitates that all tables are sourced from *actual* scientific papers. Removing or replacing these tables would violate this principle. Moreover, revising tables would also be technically challenging: if done by models, an understanding of table semantics would be needed; if done manually, the process would be highly labor-intensive. Therefore, we adopted our current approach, where LLMs generate candidate claims that are subsequently filtered by human annotators. This allows us to achieve a balance between the scale (with the help of LLMs) and the high quality (through human verification) of the dataset.
>
> Last, we agree with the reviewer that human labor is quite precious and we need to save it as long as we can. However, we found that human labor plays a quite important role in each part of our annotation pipeline, especially for handling NEI claims (Please refer to our previous response to point A"analysis of the best practice of using human labor in dataset construction." )
>
>
>
> > *QB: As in the paper, most reasoning functions in table reasoning are simple, like add, subtraction, lookup, etc. However, when these functions are used together, the results of LLMs turn out to be poor. I would like to see some analysis of the compositionality problem in table-based reasoning using this dataset.*
>
> Thanks for pointing this out. We agree that studying the compositionality problem in table-based reasoning is an interesting research direction. This study typically requires a breakdown performance evaluation and analysis for each level of reasoning depth and each type of reasoning graph. However, as we acknowledged in the Limitations section, the ground-truth reasoning depths and reasoning graphs were not included in our dataset. Annotating them would be time-consuming and costly. Given our limited time, we are unable to undertake such extensive and detailed annotation at this moment. But we do acknowledge this as an interesting future work. One possible approach is to follow the methodology in Self-Ask [2], in which the authors proposed a new metric called the "compositionality gap" to measure the capability of LLMs in compositional reasoning. However, their metric still requires the ground-truth annotation of the composed question together with its decomposed atomic questions.
>
>
> > *QC: Have you tried other advanced prompting techniques for reasoning, such as self-consistency, least-to-most prompting, etc.?*
>
> Thanks for pointing out an interesting discussion question. We have included an additional experiment for “self-consistency” prompting in response to this feedback. We use the 5-way self-consistency (the final answer is based on the majority vote of 5 generated candidate
>  answers) and test with InstructGPT and GPT-4 in the 2-class classification task. We observe that self-consistency prompting slightly improves the performance over all four different settings, with an average improvement of 2.1%. We will include this additional result in the camera-ready version.
>
> |                         | w/o self-consistency | w/ self-consistency |
> | ----------------------- | -------------------- | ------------------- |
> | InstructGPT (zero-shot) | 68.44                | **69.80**           |
> | InstructGPT + CoT       | 68.46                | **68.85**           |
> | GPT-4  (zero-shot)      | 78.22                | **78.72**           |
> | GPT-4 + CoT             | 76.85                | **80.91**           |
>
>
>
>
> > *QD: Can the human-machine collaboration pipeline be optimized? Are there any guidelines for deciding which parts of the data construction process should use human labor?*
>
> Thanks for pointing out an interesting discussion question. Our final data creation pipeline undergoes repetitive testing and revision until it reaches its current form. In our pilot annotation, we carefully evaluate the necessity and extent of human involvement (Please refer to our previous response to point A "analysis of the best practice of using human labor in dataset construction.") to optimize the human-machine collaboration pipeline. We believe our current pipeline is a balance between the scale (with the help of LLMs) and the quality (through human verification) of the dataset.
>
> With regard to the guidelines for involving human labor, we believe there is no unified guideline that fits all data annotations. We need to conduct pilot studies to determine the best strategy for human-machine collaboration on a case-by-case basis. However, we found that the principles adopted in WANLI [3] serve as a representative work for using human-machine collaboration. In this approach, data examples are automatically generated by GPT-3 and then revised and labeled by human crowd workers.
>
>
>
> > *QE: Given the fact that 20% of the errors are caused by miscalculations, I wonder if equipping the models with external tools (like calculators) will help. Parsing code from the text for the Python interpreter may be too heavy for simple calculations.*
>
> Thanks for pointing out an interesting discussion question. We agree that equipping models with calculators could potentially improve the model, and this is something we plan to explore in future work. To this end, we are considering the use of GPT-4 plugins or adopting approaches from other tool-augmented LLMs like Toolformer [4] and Chameleon [5].
>
>
>
> > *Typos Grammar Style And Presentation Improvements:I suggest using different terms between "decoder-only LLMs" and "API-based LLMs", since some API-based LLMs are also decoder-only LLMs (e.g., InstructGPT). Maybe it's better to distinguish them with "open-sourced" and "closed-sourced" models.*
>
> Thanks for the suggestions! We will revise the terms "open-sourced" and "closed-sourced" models in the final version as suggested.
>
> In conclusion, we deeply appreciate the reviewer's detailed comments and hope that our detailed response and commitment to improvement have addressed the concerns raised.
>
> [1]  Muru Zhang, Ofir Press, William Merrill, Alisa Liu, Noah A. Smith. (2023). How Language Model Hallucinations Can Snowball. https://arxiv.org/abs/2305.13534
>
> [2] Ofir Press, Muru Zhang, Sewon Min, Ludwig Schmidt, Noah A. Smith, Mike Lewis. (2022). Measuring and Narrowing the Compositionality Gap in Language Models.  https://arxiv.org/abs/2210.03350
>
> [3] Alisa Liu, Swabha Swayamdipta, Noah A. Smith, Yejin Choi (2022). WANLI: Worker and AI Collaboration for Natural Language Inference Dataset Creation. In Proceedings of Empirical Methods in Natural Language Processing (EMNLP). https://aclanthology.org/2022.findings-emnlp.508.pdf
>
> [4] Timo Schick, Jane Dwivedi-Yu, Roberto Dessì, Roberta Raileanu, Maria Lomeli, Luke Zettlemoyer, Nicola Cancedda, Thomas Scialom. (2023). Toolformer: Language Models Can Teach Themselves to Use Tools. https://arxiv.org/abs/2302.04761
>
> [5] Pan Lu, Baolin Peng, Hao Cheng, Michel Galley, Kai-Wei Chang, Ying Nian Wu, Song-Chun Zhu, Jianfeng Gao. (2023). Chameleon: Plug-and-Play Compositional Reasoning with Large Language Models.  https://arxiv.org/abs/2304.09842

---

### Official Review · Reviewer_P1EZ · 2023-08-04

**Soundness:** 4

**Excitement:**

3: Ambivalent: It has merits (e.g., it reports state-of-the-art results, the idea is nice), but there are key weaknesses (e.g., it describes incremental work), and it can significantly benefit from another round of revision. However, I won't object to accepting it if my co-reviewers champion it.

**Paper Topic And Main Contributions:**

In this paper, a new dataset SCITAB is constructed for addressing some of the current shortcomings of current scientific fact-checking benchmarks. The SCITAB dataset has a high degree of difficulty, with all models except GPT-4 barely outperforming randomized guessing, and provides an excellent dataset for further subsequent research in this area.

**Reasons To Accept:**

A.Adequate experiments have been conducted in this paper and the results have been analyzed in sufficient detail.
B.All models except GPT-4 perform poorly on the dataset constructed in this paper, leaving room for further research.
C.The paper is easy to follow.

**Reasons To Reject:**

A.The atomic reasoning functions are not distributed reasonably enough, and the two types of The Simple lookup and The Comparison are over-represented.
B.The use of LLM to generate data reduces the workload, but the final dataset had significantly less data than the comparison dataset.

**Reproducibility:**

3: Could reproduce the results with some difficulty. The settings of parameters are underspecified or subjectively determined; the training/evaluation data are not widely available.

**Reviewer Confidence:**

3: Pretty sure, but there's a chance I missed something. Although I have a good feel for this area in general, I did not carefully check the paper's details, e.g., the math, experimental design, or novelty.

---

> ### Author Rebuttal · Authors · 2023-08-28
>
> We are sincerely grateful for the time Reviewer P1EZ has invested in reviewing our paper, and for the encouraging evaluation that underscores the positive aspects of our work. The insights into the construction of the SCITAB dataset, its high degree of difficulty, and its potential to spark further research are truly appreciated. The observations reflect a genuine understanding of the paper's main contributions and the reasons to accept it. We appreciate the valuable insights behind the comments and also want to clarify a few points.
>
>
>
> > *A. The atomic reasoning functions are not distributed reasonably enough, and the two types of The Simple lookup and The Comparison are over-represented.*
>
> We appreciate the reviewer for identifying an interesting discussion point regarding the distribution of atomic reasoning functions.  We double-checked the calculation of the reasoning depths statistics and we found that it's true that Simple Lookup and Comparison account for a significant part of our dataset, representing 20.6% and 19.5%. Note that during data collection, we do not enforce any specific rules to favor certain types of claims. Therefore, this distribution is not the result of cherry-picking but rather a reflection of the natural distribution of real-world scientific claims. Actually, it is understandable that Simple Lookup and Comparison are more common in scientific claims. For example, authors often describe experimental findings by either reporting specific data points from tables (e.g., "Model A has a performance of 50," which is a simple lookup) or by comparing performances between two models (e.g., "Model B is 30% higher than Model A," which falls under comparison). Therefore, we believe this distribution is quite reasonable and reflective of real-world academic practices. Lastly, we would like to clarify that even though Simple Lookup and Comparison functions are common in our dataset, this does not imply that the scientific claims in our dataset are simple. In fact, Simple Lookup and Comparison functions do not just exist in simple claims; we found that many complex claims also use Simple Lookup and Comparison because these functions often serve as the foundation for more complex reasoning.
>
>
>
> > *B. The use of LLM to generate data reduces the workload, but the final dataset had significantly less data than the comparison dataset.*
>
> We thank the reviewer for pointing this out. First, we acknowledge the relatively small size of our SCITAB dataset due to the involvement of human efforts. However, even with the LLM to help reduce the workload, human efforts are still imperative to ensure the data quality. To ensure the high-quality and challenging nature of the dataset, we employed a strict data annotation process. Many stages of our annotation pipeline involve manual verification to ensure the quality of the data, which naturally limits the final quantity. In the trade-off between quantity and quality, we prioritize quality as our goal is to create a diagnostic dataset that serves as a reliable benchmark for model evaluation. As highlighted in the paper's title and introduction, the primary purpose of this dataset is to serve as an evaluation dataset, aiming to present a challenging benchmark for compositional reasoning and claim verification on scientific tables. We believe the size of 1.2K is sufficient for an evaluation dataset as we do not include a training set. This is a comparable size with many existing evaluation datasets such as HumanEval [1] (164 samples) and SciBench [2] (695 samples).
>
> In conclusion, we deeply appreciate the reviewer's constructive comments and hope that our detailed response and commitment to improvement have addressed the concerns raised.
>
> [1] Chen et al. Evaluating Large Language Models Trained on Code. (2021). https://arxiv.org/abs/2107.03374
>
> [2] Wang et al. SciBench: Evaluating College-Level Scientific Problem-Solving Abilities of Large Language Models. (2023). https://arxiv.org/abs/2307.10635

---

### Official Review · Reviewer_z4dP · 2023-08-05

**Soundness:** 4

**Excitement:**

3: Ambivalent: It has merits (e.g., it reports state-of-the-art results, the idea is nice), but there are key weaknesses (e.g., it describes incremental work), and it can significantly benefit from another round of revision. However, I won't object to accepting it if my co-reviewers champion it.

**Paper Topic And Main Contributions:**

The paper introduces the SciTab dataset, which consists of tabular data paired with claims that can be labeled "supported", "refuted" or "not enough information" in the context of the table. The dataset starts from the existing SciGen dataset, filtering down to "check-worthy" claims. The InstructGPT is used to construct a claim with opposite meaning. InstructGPT is also used to construct other claims for the table. All claims are then verified by human annotators.

In total the dataset has 1225 claims fairly evenly spread over the 3 classes, with about 6 claims per table.

Analysis of the dataset shows a diverse set of reasoning types (like "comparison" and "max/min"), as well as a large fraction of claims requiring 3 or more "reasoning steps".

Various few-shot LLM and table-trained models are evaluated, including GPT-4 which reaches 3-class macro F1 scores of ~65, indicating that there's still some challenges in the data (although not clear what a human baseline would be). Smaller models generally do quite poorly, further indicating this is a hard task. A "program-of-thought" also doesn't perform very well.

There's a good amount of error analysis, with explicit examples given in the appendix.

**Reasons To Accept:**

A new dataset on a potentially impactful task of verifying claims back by tabular data.

Analysis showing diversity of data. Interesting evaluations, showing it's a challenging task, with GPT-4 having the best performance. Detailed error analysis.

**Reasons To Reject:**

Judging from the examples in the paper, it's unclear how well defined the task is. E.g., the main example in Figure 1 relies on external "commensense knowledge" that "The number of random chance is 50%" which doesn't seem clear from the context, as far as I can tell?  (also saying that 57.5% is 7.5% more than 50% is a bit unusual, at least ambiguous). On the error analysis of InstructGPT in Figure 11, two of the examples are predicted as NEI which might be understandable since the claim mentions "Portuguese" while the table only has "PT".

It would be nice with a clearer human baseline, beyond the inter-annotator agreement indirect number (cohen kappa of 0.64 between two annotators).

**Reproducibility:**

4: Could mostly reproduce the results, but there may be some variation because of sample variance or minor variations in their interpretation of the protocol or method.

**Reviewer Confidence:**

4: Quite sure. I tried to check the important points carefully. It's unlikely, though conceivable, that I missed something that should affect my ratings.

---

> ### Author Rebuttal · Authors · 2023-08-28
>
> We thank Reviewer z4dP for the thoughtful feedback. First, we are encouraged that Reviewer z4dp recognized the novelty of our work in introducing the SCITAB dataset, which offers a unique and significant task of verifying claims backed by tabular data, highlighting diverse reasoning types. Also, we are delighted that the reviewer confirms the challenges in the data, and the detailed error analysis, all of which add significant value to the research community.
>
> We appreciate the valuable insights behind the comments and also want to clarify a few points.
>
>
> > *A. Judging from the examples in the paper, it's unclear how well-defined the task is.*
>
> Our task definition is to specifically verify the claim based on the table information provided, and this was emphasized in the annotation training guidelines. For example, we emphasized that the claim has to be exactly supported/ refuted by the table information, and we provided examples to the annotators during training on what should not be included. While the task definition is covered in Section 4 (Lines 318 - 328), the training process for annotators regarding task clarification was not included in the paper. We thank the reviewer for pointing this out. We will include the detailed task definition for training the annotators in the current Appendix B with a new subsection "Annotator training process" and append all the guidelines.
>
>
>
> > *B. E.g., the main example in Figure 1 relies on external "commonsense knowledge" that "The number of random chance is 50%".*
> >
> > *On the error analysis of InstructGPT in Figure 11, two of the examples are predicted as NEI which might be understandable since the claim mentions "Portuguese" while the table only has "PT"*.
>
> We appreciate the reviewer's detailed observation. We note that certain commonsense knowledge, like understanding that a random chance is 50%, is naturally occurring in human reasoning. For the other example in Figure 11, the table contains only the abbreviations 'EN' and 'PT.' Since 'EN' represents English, our annotators logically deduced that 'PT' stands for Portuguese.  Similar to WANLI [1], during our training sessions, such commonsense interpretations and a minimum amount of logical inference are acceptable. Therefore, we believe that our examples are aligned with the task definition, although the interpretation of "minimum amount of logical inference and commonsense knowledge" can vary for different people.
>
> We view this phenomenon as an interesting discussion point, rather than a definitive issue in our dataset. We will explain this in detail in Appendix B of the camera-ready version.
>
>
>
> > *C. It would be nice with a clearer human baseline, beyond the inter-annotator agreement indirect number (cohen kappa of 0.64 between two annotators).*
>
> We appreciate the reviewer's suggestion on this. To examine how humans perform on our SCITAB dataset, we hired an annotator from our candidate annotators pool, following the same training procedure as other annotators.
>
> In the case of 2-class classification, we randomly selected 40 samples—20 each for supported and refuted claims. For 3-class classification, we randomly selected 60 random samples, ensuring an even distribution of 20 samples across the three label categories (supported, refuted, and not enough information). The annotator took approximately 1.5 hours for the 2-class task and 2 hours for the 3-class task. We paid 11.2 USD per hour at the same rate described in the paper.
>
> We report the Macro-F1 scores for human performance, together with the corresponding GPT-4 performance for comparison, as shown in the Table below (to be added in Section 4). We observe that a well-trained human annotator can achieve 92.46 and 84.73 F1 scores on the 2-class and 3-class fact-checking, respectively. GPT-4, while performing the best among all models, still represents a large gap from human performance. We will include these new results in the main results table (Table 4) and the above findings in Section 4.2.
>
>
> |       | 2-class | 3-class |
> | ----- | ------- | ------- |
> | GPT-4 | 77.98   | 63.21   |
> | Human | **92.46**   | **84.73**   |
>
>
>
> In conclusion, we deeply appreciate the reviewer's constructive comments and hope that our detailed response and commitment to improvement have addressed the concerns raised.
>
> [1] Alisa Liu, Swabha Swayamdipta, Noah A. Smith, Yejin Choi (2022). WANLI: Worker and AI Collaboration for Natural Language Inference Dataset Creation. In Proceedings of Empirical Methods in Natural Language Processing (EMNLP). https://aclanthology.org/2022.findings-emnlp.508.pdf

---

### Meta-Review · Area_Chair_U5iu · 2023-09-19

**Recommendation:** 4

**Metareview:**

Summary: This paper introduces a new dataset called SCITAB, which is designed to address the limitations of existing benchmarks for scientific fact-checking. The dataset consists of 1225 challenging scientific claims that require compositional reasoning with scientific tables. These claims are derived from actual scientific statements, and the evidence is presented in the form of tables, which mirror real-world fact-checking scenarios. The authors have established benchmarks on SCITAB using state-of-the-art models and have identified unique challenges, such as ambiguous expressions and irrelevant claims.

Strengths: The main contribution of this paper is the creation of the SCITAB dataset. All the reviewers unanimously agree that this is a solid contribution and provides a new resource for the research community to further explore and improve scientific fact-checking. The paper is well-written and easy to follow.

Weaknesses: I don't think there are any major weaknesses -- some of the weaknesses have been addressed during the discussion phase. The authors should take the experimental results during the rebuttals into account when preparing for the final version of the paper.

---

### Decision · Program_Chairs · 2023-10-07

**Decision:**

Accept-Main

**Comment:**

Summary: This paper introduces a new dataset called SCITAB, which is designed to address the limitations of existing benchmarks for scientific fact-checking. The dataset consists of 1225 challenging scientific claims that require compositional reasoning with scientific tables. These claims are derived from actual scientific statements, and the evidence is presented in the form of tables, which mirror real-world fact-checking scenarios. The authors have established benchmarks on SCITAB using state-of-the-art models and have identified unique challenges, such as ambiguous expressions and irrelevant claims.

Strengths: The main contribution of this paper is the creation of the SCITAB dataset. All the reviewers unanimously agree that this is a solid contribution and provides a new resource for the research community to further explore and improve scientific fact-checking. The paper is well-written and easy to follow.

Weaknesses: I don't think there are any major weaknesses -- some of the weaknesses have been addressed during the discussion phase. The authors should take the experimental results during the rebuttals into account when preparing for the final version of the paper.